# Oxidized LDL-dependent pathway as new pathogenic trigger in arrhythmogenic cardiomyopathy

Elena Sommariva[1],[*],[†] iD, Ilaria Stadiotti[1],[†] iD, Michela Casella[2], Valentina Catto[2] iD, Antonio Dello Russo[2], Corrado Carbucicchio[2], Lorenzo Arnaboldi[3] iD, Simona De Metrio[3], Giuseppina Milano[1],[4], Alessandro Scopece[1], Manuel Casaburo[1], Daniele Andreini[5],[6], Saima Mushtaq[5], Edoardo Conte[5], Mattia Chiesa[7] iD, Walter Birchmeier[8], Elisa Cogliati[9], Adolfo Paolin[9], Eva König[10] iD, Viviana Meraviglia[10] iD, Monica De Musso[10], Chiara Volani[10], Giada Cattelan[10] iD, Werner Rauhe[11], Linda Turnu[12] iD, Benedetta Porro[12], Matteo Pedrazzini[13] iD, Marina Camera[3],[14], Alberto Corsini[3],[15] iD, Claudio Tondo[2],[16], Alessandra Rossini[10] & Giulio Pompilio[1],[16]

## Abstract

**Arrhythmogenic cardiomyopathy (ACM) is hallmarked by ventricular fibro-adipogenic alterations, contributing to cardiac dysfunctions and arrhythmias. Although genetically determined (e.g., *PKP2* mutations), ACM phenotypes are highly variable. More data on phenotype modulators, clinical prognosticators, and etiological therapies are awaited. We hypothesized that oxidized low-density lipoprotein (oxLDL)-dependent activation of PPARγ, a recognized effector of ACM adipogenesis, contributes to disease pathogenesis. ACM patients showing high plasma concentration of oxLDL display severe clinical phenotypes in terms of fat infiltration, ventricular dysfunction, and major arrhythmic event risk. In ACM patient-derived cardiac cells, we demonstrated that oxLDLs are major cofactors of adipogenesis. Mechanistically, the increased lipid accumulation is mediated by oxLDL cell internalization through CD36, ultimately resulting in PPARγ upregulation. By boosting oxLDL in a *Pkp2* heterozygous knock-out mice through high-fat diet feeding, we confirmed *in vivo* the oxidized lipid dependency of cardiac adipogenesis and right ventricle systolic impairment, which are counteracted by atorvastatin treatment. The modulatory role of oxidized lipids on ACM adipogenesis, demonstrated at cellular, mouse, and patient levels, represents a novel risk stratification tool and a target for ACM pharmacological strategies.**

**Keywords** Arrhythmogenic Cardiomyopathy; ARVC; adipogenesis; oxidative stress; lipoproteins

**Subject Category** Cardiovascular System

## Introduction

Arrhythmogenic cardiomyopathy (ACM) is a heart condition with genetic traits. It is hallmarked by a gradual fibro-adipose

1   Unit of Vascular Biology and Regenerative Medicine, Centro Cardiologico Monzino IRCCS, Milan, Italy
2   Heart Rhythm Center, Centro Cardiologico Monzino IRCCS, Milan, Italy
3   Department of Pharmacological and Biomolecular Sciences, Università degli Studi di Milano, Milan, Italy
4   Department of Heart and Vessels, Laboratory of Cardiovascular Research, University Hospital of Lausanne, Lausanne, Switzerland
5   Unit of Cardiovascular Imaging, Centro Cardiologico Monzino IRCCS, Milan, Italy
6   Department of Clinical Sciences and Community Health, Università degli Studi di Milano, Milan, Italy
7   Bioinformatics and Artificial Intelligence facility, Centro Cardiologico Monzino IRCCS, Milan, Italy
8   Max Delbrück Center for Molecular Medicine, Berlin-Buch, Germany
9   Treviso Tissue Bank Foundation, Treviso, Italy
10  Institute for Biomedicine, Eurac Research, Affiliated Institute of the University of Lübeck, Bozen, Italy
11  General Hospital Bolzano/Bozen, Bolzano, Italy
12  Unit of Metabolomics and Cellular Biochemistry of Atherothrombosis, Centro Cardiologico Monzino IRCCS, Milan, Italy
13  Laboratory of Cardiovascular Genetics, Istituto Auxologico Italiano, IRCCS, Milan, Italy
14  Unit of Cell and Molecular Biology in Cardiovascular Diseases, Centro Cardiologico Monzino IRCCS, Milan, Italy
15  IRCCS MultiMedica, Milan, Italy
16  Department of Biomedical, Surgical and Dental Sciences, Università degli Studi di Milano, Milan, Italy
    *Corresponding author. Tel: +39 02 58002026; Fax: +39 0258002342; E-mail: elena.sommariva@cardiologicomonzino.it
    †These authors contributed equally to this work

replacement of the ventricular myocardium, heart failure, malignant arrhythmias, and sudden death (Basso *et al*, 2009).

ACM inheritance has mainly an autosomal dominant pattern and most causative mutations reside in desmosomal genes, especially *PKP2* (Lazzarini *et al*, 2015). The reasons underpinning the characteristic low penetrance and variable expressivity (Pinamonti *et al*, 2014) are still not properly understood. Genetic determinants and environmental factors could play a role as phenotypic modulators (Xu *et al*, 2010; König *et al*, 2017). For instance, physical exercise worsens both ACM arrhythmic burden (La Gerche, 2015) and myocardial dysfunction (Saberniak *et al*, 2014).

Disease severity markers must be improved, and only few pharmacological approaches to counteract disease progression have been proposed (Chelko *et al*, 2016; Chelko *et al*, 2019).

ACM electrical dysfunction is determined by genetically impaired cardiomyocytes and worsened by adipose tissue presence (Samanta *et al*, 2016; De Coster *et al*, 2018), which we showed to derive, at least partly, from differentiation of cardiac mesenchymal stromal cells (C-MSCs; (Lombardi *et al*, 2016; Sommariva *et al*, 2016; Stadiotti *et al*, 2017)). C-MSC, being primary human cells directly obtained from patients ventricular biopsy, easy to isolate and to amplify (Pilato *et al*, 2018), carrying the whole patient's genetic setting and recapitulating ACM adipogenesis *in vitro* (Sommariva *et al*, 2016), represent a valid cell model to study ACM.

A key regulator of ACM adipogenesis is the proliferator-activated receptor gamma (PPARγ), whose activation is dependent on Wnt/βcatenin- (Garcia-Gras *et al*, 2006) and Hippo-pathway (Chen *et al*, 2014) impairment. Accordingly, PPARγ modulators rosiglitazone or 13-hydroxy-octadecadienoic acid (13HODE) have been used to shift glycolysis to fatty acid metabolism to model ACM lipogenesis in human induced pluripotent stem cell (hiPSC)-derived cardiomyocytes (hiPSC-CM) (Kim *et al*, 2013). In other conditions such as atherosclerosis, the involvement of oxidized low-density lipoproteins (oxLDL) and their component 13HODE in regulating cell lipid accumulation is well known. In particular, after oxLDL cellular internalization by scavenger receptors, such as CD36 (Sun *et al*, 2007), 13HODE directly induces PPARγ expression (Tontonoz *et al*, 1998), provoking CD36 expression increase, thus facilitating additional oxLDL uptake and further PPARγ activation in a feed-forward circle (Jostarndt *et al*, 2004). Notably, 13HODE blood concentrations have been used as an exercise-induced oxidative stress marker in athletes (Powers & Jackson, 2008; Nieman *et al*, 2014; Sugama *et al*, 2015).

An adjunctive clue hinting to a link between LDL levels and cardiac adipocyte accumulation comes from available ACM-transgenic mice, which fail to accumulate considerable amount of myocardial fat (McCauley & Wehrens, 2009). In fact, mice, unlike humans, show low circulating total cholesterol, which mostly resides in HDL fractions (Kashyap *et al*, 1995).

We thus hypothesized that oxLDL/CD36/PPARγ circuitry may be a trigger of adipogenesis, and related clinical phenotypes, in ACM. Through the investigation of ACM patient features, *in vitro* experiments on patient-derived cardiac cells (both C-MSC and hiPSC-CM) and *in vivo* studies with *Pkp2* heterozygous knock-out mice (*Pkp2*+/−), we consistently showed for the first time that oxLDL and elevated oxidative status contribute to ACM phenotype severity.

## Results

### ACM patients show high plasma and cardiac lipid peroxidation indices

We evaluated oxLDL and 13HODE plasma concentrations in ACM patients and HC, matched for sex, age, and cardiovascular risk factors (Appendix Table S1). As reported in Fig 1A, ACM patients showed higher plasma oxLDL compared to HC ($n = 36$; ACM $137.90 \pm 20.85$ vs. HC $66.74 \pm 5.79$ ng/ml; $P = 0.015$). Accordingly, plasma 13HODE was found significantly higher in patients than controls (Appendix Table S1).

Interestingly, we observed a difference in oxLDL mean levels between patients with overt ACM phenotype carrying ACM-related causative mutations and their relatives, carriers of the same mutations but not clinically affected by the pathology ($n = 7$ vs. $n = 9$; ACM $384.50 \pm 139.1$ vs. unaffected relatives $66.99 \pm 17.09$ ng/ml; $P = 0.02$; Fig 1B; Appendix Fig S1; Appendix Table S2). This evidence points to an association between a fully penetrant disease and high oxLDL plasma concentration.

In addition, we did not observe differences in oxLDL plasma concentrations between patients with *PKP2* mutations and patients with mutations in other ACM-associated genes or gene elusive ($n = 12$ vs. $n = 22$; Fig 1C; Appendix Table S3; Appendix Fig S2; Appendix Table S5), indicating that oxLDL elevation is not linked only to *PKP2* forms.

To understand whether high plasma oxLDL and 13HODE have a correspondence at cardiac tissue levels, we quantified the lipid peroxidation marker MDA on ACM and control RV sections, finding higher oxidative stress in ACM hearts ($n = 4$; MDA relative densitometric analysis (d.a.)/nuclei numbers ACM $36.25 \pm 10.43$ vs. HC $1.00 \pm 0.72$; $P = 0.015$; Fig 1D).

CD36 receptor has a central role in oxLDL uptake (Tontonoz *et al*, 1998). CD36 immunostaining on RV tissue from ACM and HC donors revealed higher CD36 expression in ACM samples ($n = 4$; CD36 relative d.a./nuclei numbers ACM $14.72 \pm 2.10$ vs. HC $1.00 \pm 0.40$; $P = 0.0007$; Fig 1E), which was mainly distributed in replacement tissue areas.

### Elevated oxLDL plasma concentrations are associated with structural and functional impairment and arrhythmic burden in ACM patients

Since Fig 1B suggested a role of oxLDL in ACM phenotype worsening, we retrospectively investigated the association between plasma oxLDL levels and structural, functional, and arrhythmic features in our whole ACM patient cohort. A ROC curve analysis identified the cut-off value of 86 ng/ml which best discriminates ACM patients vs. HC ($n = 36$; 63.41% sensitivity and 65.85% specificity; Fig 2A). Based on this cut-off value, we subdivided our ACM patient cohort in two groups ($n = 26 < 86$ ng/ml oxLDL and $n = 41 > 86$ ng/ml oxLDL). In the sub-cohort of ACM patients for which MRI was performed in our hospital, we quantified the mass of ventricular fat infiltration (Aquaro *et al*, 2014). Strikingly, patients with oxLDL plasma concentrations above the cut-off showed significantly higher myocardial fat infiltration ($n = 14$ vs. $n = 25$; fat infiltration mass $< $ oxLDL $2.27 \pm 1.35$ vs. $> $ oxLDL $15.32 \pm 4.62$ grams; $P = 0.04$; Fig 2B). Beside greater structural impairment, the group with higher

oxLDL also showed a higher frequency of RV dysfunction defined as in (Marcus *et al*, 2010) (n = 26 vs. n = 41; % of patients with RV dysfunction < oxLDL 26.9% (7/26) vs. > oxLDL 53.7% (22/41); P = 0.04; Fig 2C), biventricular dysfunction (n = 26 vs. n = 41; % of patients with biventricular dysfunction < oxLDL 0% (0/26) vs.

> oxLDL 19.5% (8/41); P = 0.02; Fig 2D), and RV wall motion abnormalities (n = 26 vs. n = 41; % RV wall motion abnormalities < oxLDL 38.4% (19/26) vs. > oxLDL 70.7% (29/41); P = 0.01; Fig 2E). Accordingly, RV EF was significantly reduced in patients with higher levels of oxLDL (n = 23 vs. n = 39; % RV EF < oxLDL

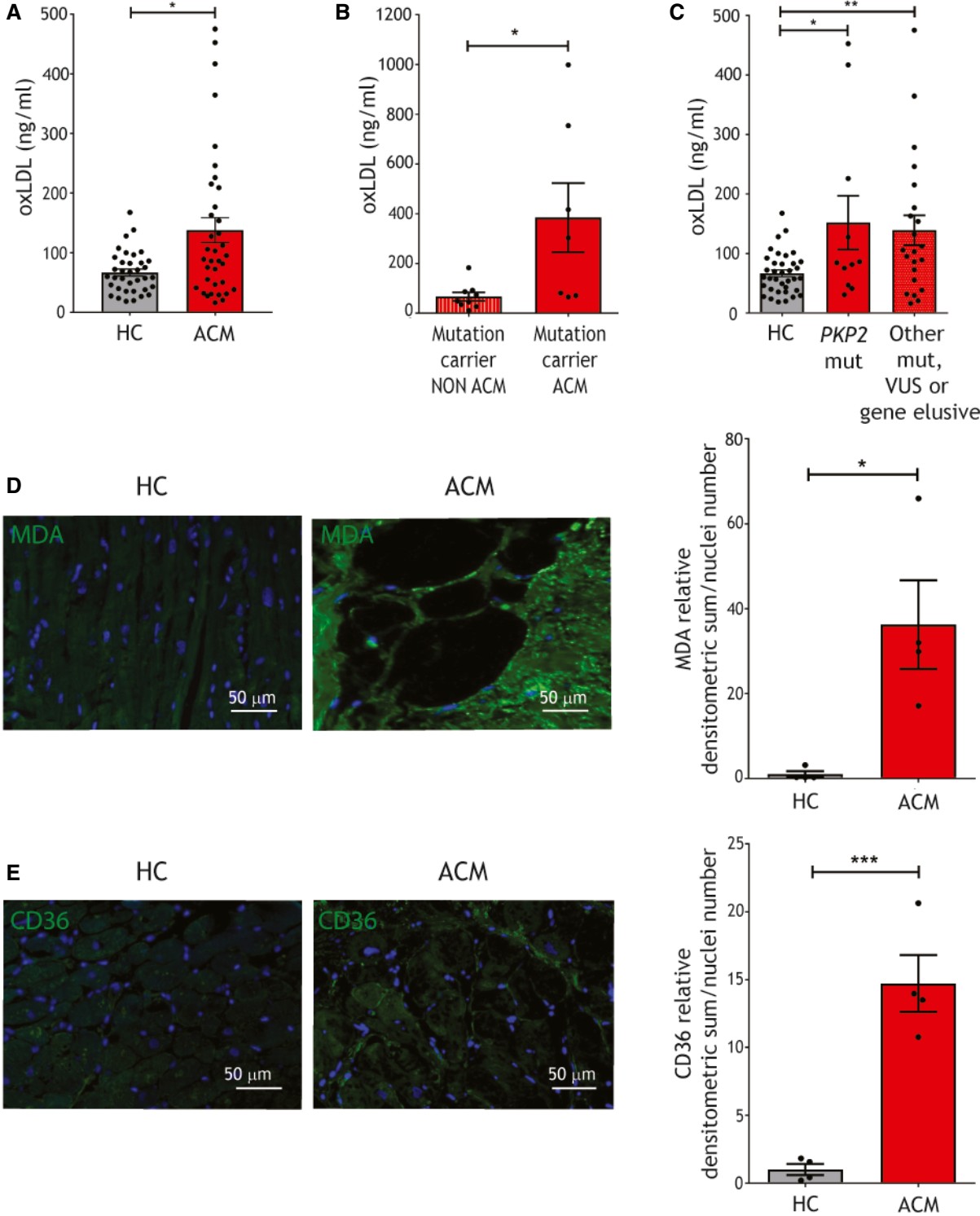

**Figure 1.**

**Figure 1. ACM patients show high plasma oxLDL, cardiac lipid peroxidation, and oxLDL receptor levels.**

A   oxLDL plasma concentration in ACM patients and HC ($n$ = 36; Mann–Whitney test).
B   oxLDL plasma concentration in mutated ACM ($n$ = 7) and non-ACM relatives, carriers of the same causative mutation ($n$ = 9; Mann–Whitney test).
C   oxLDL plasma concentration in ACM patients carriers of a *PKP2* mutation and ACM patients carriers of other desmosomal or non desmosomal mutations, or gene elusive ($n$ = 10 vs. $n$ = 26; one-way ANOVA).
D   Representative images of MDA immunostaining (green) on ACM and HC ventricular tissue sections and relative quantification ($n$ = 4 biological replicates; two-tailed Student's *t*-test). Nuclei are counterstained with Hoechst33342 (blue).
E   Representative images of CD36 immunostaining (green) on ACM and HC ventricular tissue sections and relative quantification ($n$ = 4 biological replicates; two-tailed Student's *t*-test). Nuclei are counterstained with Hoechst33342 (blue).

Data information: mean ± SEM. *$P$ < 0.05; **$P$ < 0.01; and ***$P$ < 0.001.
Source data are available online for this figure.

50.71 ± 2.40 vs. > oxLDL 44.5 ± 1.72; $P$ = 0.04; Fig 2F), confirming that higher oxLDL is associated with functional impairment. Intriguingly, a Kaplan–Meier analysis revealed that ACM patient survival free from major arrhythmic events (MAEs) over 5-year follow-up was significantly higher in the subset of patients with low amount of oxLDL ($n$ = 67; log-rank $P$ < 0.0001; hazard ratio = 0.223 [0.116–0.428]; mean follow-up 7.99 ± 0.39 years; Fig 2G), demonstrating the association between oxLDL levels and the occurrence of MAE in ACM patients.

## ACM C-MSC show elevated oxidative stress and PPARγ and CD36 expression

To model adipogenesis *in vitro*, we obtained ACM and HC C-MSC from RV endomyocardial biopsies (Casella *et al*, 2020), as previously described (Pilato *et al*, 2018). To evaluate ACM C-MSC reactive oxygen species (ROS), we performed a dichlorofluorescein (DCF) test. As shown in Fig 3A, oxidative stress is significantly higher in ACM C-MSC compared to HC C-MSC ($n$ = 5; relative mean DCF emission ACM 1.63 ± 0.26 vs. HC 1.00 ± 0.06; $P$ = 0.049). At least part of the ACM C-MSC oxidative stress resulted in lipid peroxidation, as shown by MDA cell immunofluorescence ($n$ = 4 vs. $n$ = 5; MDA relative d.a./nuclei number ACM 8.83 ± 2.78 vs. HC 1.00 ± 0.51; $P$ = 0.017; Fig 3B). To understand whether excess oxidative stress is due to a defect in antioxidant capacity, we measured the ratio between reduced (GSH) and oxidized glutathione levels in HC and ACM C-MSC, obtaining no differences between the two groups ($n$ = 8; Fig 3C).

In addition, ACM C-MSC in growth medium (GM) showed higher PPARγ and CD36 levels compared to HC ($n$ = 18; PPARγ/GAPDH d.a. ACM 1.47 ± 0.12 vs. HC 1.00 ± 0.06; $P$ = 0.002; $n$ = 17 vs. $n$ = 16; CD36/GAPDH d.a. ACM 1.52 ± 0.20 vs. HC 1.00 ± 0.16; $P$ = 0.02; Fig 3D).

## Lipid accumulation and CD36 levels increase in ACM C-MSC upon adipogenic stimulus

Since corresponding transcription levels of CD36 and PPARγ, as well as lipid accumulation, are described for other cell types (Tontonoz *et al*, 1998), we evaluated whether ACM C-MSC, already predisposed to PPARγ activation (Sommariva *et al*, 2016), were more prone to expose CD36 on the plasma membrane during lipid accumulation. We performed a double staining with Nile Red, marking neutral lipids, and anti-CD36 antibody in ACM and HC C-MSC, cultured in adipogenic medium (AM) for different time-points (0, 3,

6, and 10 days). Figure 3E shows that, during adipogenic differentiation, ACM C-MSC simultaneously increase CD36 and lipid content significantly more than HC cells. In both ACM and HC cells, a linear correlation is present between these two parameters ($P$ = 0.008 and 0.03, respectively); however, a higher slope in ACM C-MSC was evident ($n$ = 4 vs. $n$ = 5; ACM slope 1.33; $R^2$ = 0.99 vs. HC slope 0.69; $R^2$ = 0.93; slopes statistically different $P$ = 0.016). To explore CD36 activity during adipogenic differentiation, we evaluated C-MSC oxLDL internalization, by assessing the intracellular fluorescence after 10 μg/ml DiI dye-conjugated oxLDL treatment (DiI-oxLDL), either in GM or in AM. ACM C-MSC internalized more DiI-oxLDL in AM compared to GM ($n$ = 3; DiI internalization ACM GM 2.22 ± 0.16 vs. ACM AM 6.29 ± 0.65; $P$ = 0.002) and 3-fold more than controls in AM ($P$ = 0.01; Fig 3F). Furthermore, lipidomic assays confirmed ACM C-MSC transition toward an adipogenic lineage (Appendix Fig S3).

## OxLDL and 13HODE enhance lipid accumulation in ACM C-MSC

To test the hypothesis that oxLDL exacerbate adipogenic propensity in ACM C-MSC through a vicious circle implicating CD36 and PPARγ, we treated ACM and HC C-MSC with or without 150 μg/μl oxLDL in AM. The presence of oxLDL caused increased lipid accumulation in ACM C-MSC compared to AM only ($n$ = 11; ORO relative lipid accumulation ACM AM 6.19 ± 0.83 vs. ACM AM+oxLDL 11.86 ± 2.64; $P$ = 0.01; Fig 4A), as well as CD36 and PPARγ expression ($n$ = 5; PPARγ/GAPDH d.a. ACM AM 1.52 ± 0.13 vs. ACM AM+oxLDL 2.78 ± 0.44; $P$ = 0.01; $n$ = 5; CD36/GAPDH d.a. ACM AM 1.18 ± 0.61 vs. ACM AM+oxLDL 2.29 ± 0.95; $P$ = 0.01; Fig 4A).

By mimicking *PKP2* haploinsufficiency through a silencing approach, we demonstrated that oxLDL susceptibility is dependent on desmosomal protein defect (Appendix Fig S4).

To understand whether 13HODE is one of the determinants of oxLDL-dependent adipogenesis in ACM, we cultured ACM and HC C-MSC for 72 h in AM, with or without 20 μg/ml 13HODE. Importantly, 13HODE significantly increased lipogenesis in ACM C-MSC only ($n$ = 12; ORO relative lipid accumulation ACM AM 3.60 ± 0.82 vs. ACM AM+13HODE 4.75 ± 0.92; $P$ = 0.048; Fig 4B). In the same experiment, we obtained PPARγ and CD36 level upregulation after 13HODE treatment in ACM C-MSC ($n$ = 8; PPARγ/GAPDH d.a. ACM AM 1.67 ± 0.16 vs. ACM AM+13HODE 2.17 ± 0.20; $P$ = 0.047; $n$ = 8; CD36/GAPDH d.a. ACM AM 1.22 ± 0.06 vs. ACM AM+13HODE 1.78 ± 0.20; $P$ = 0.008; Fig 4C), as expected in light of the previously described mechanism (Jostarndt *et al*, 2004).

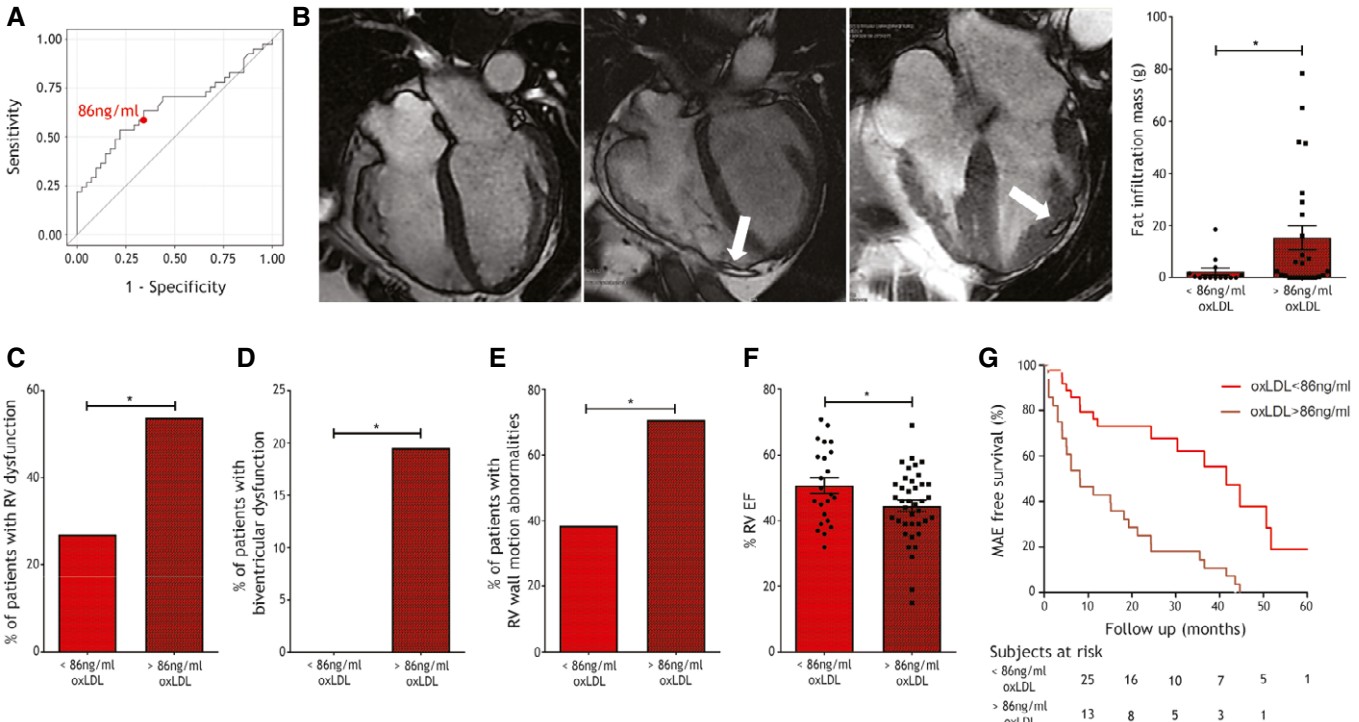

**Figure 2.  Plasma oxLDL cut-off value of 86 ng/ml defines two patient subpopulations with different severity of ACM phenotypes.**

A  ROC curve shows the capability of plasma oxLDL concentration in discriminating ACM vs. HC subjects (data as in Fig 1A). The red dot shows the point where the difference between sensitivity and specificity is minimized; this value, corresponding to the oxLDL cut-off = 86 ng/ml, was used to part the whole ACM cohort in two subpopulations.

B  Left panels: representative images of three cases of ACM in steady-state free procession sequences at cardiac MRI. On the left, a case with RV dilation, regional bulging of the RV wall without fat infiltration, and oxLDL levels below the cut-off; in the center and on the right, two cases with fat infiltration in the right or left ventricle wall (white arrow), respectively, and oxLDL levels above the cut-off. Right panel: quantification of the myocardial fat mass of the two ACM subpopulations (above or below the cut-off) whose MRI was available for re-analysis ($n = 14$ oxLDL < 86 ng/ml, $n = 25$ oxLDL > 86 ng/ml; two-tailed Student's $t$-test).

C  Frequency of patients showing RV dysfunction (defined as in (31)) in the two ACM cohort subgroups ($n = 26$ oxLDL < 86 ng/ml vs. $n = 41$ oxLDL > 86 ng/ml; two-tailed Student's $t$-test).

D  Frequency of patients showing biventricular dysfunction in the two ACM cohort subgroups ($n = 26$ oxLDL < 86 ng/ml vs. $n = 41$ oxLDL > 86 ng/ml; two-tailed Student's $t$-test).

E  Frequency of patients showing RV wall motion abnormalities in the two ACM cohort subgroups ($n = 26$ oxLDL < 86 ng/ml vs. $n = 41$ oxLDL > 86 ng/ml; Fisher's exact test).

F  RV ejection fraction % of the patients classified in the two ACM cohort subgroups ($n = 23$ oxLDL < 86 ng/ml vs. $n = 39$ oxLDL > 86 ng/ml; two-tailed Student's $t$-test).

G  Kaplan–Meier analysis of actual MAE-free survival of patient belonging to the two ACM cohort subgroups in the first 5-year follow-up ($n = 26$ oxLDL < 86 ng/ml vs. $n = 41$ oxLDL > 86 ng/ml; log-rank $P < 0.0001$; HR = 0.223[0.116–0.428]).

Data information: mean ± SEM. *$P < 0.05$.
Source data are available online for this figure.

## The antioxidant compound NAC reduces lipid accumulation in ACM C-MSC

In an attempt to prevent the effects of oxidized agents on lipid droplet accumulation in ACM cells, we tested the effect of the antioxidant NAC. 5 mmol/l NAC treatment was able to contain 13HODE-dependent lipid accumulation in ACM cells ($n = 12$; ORO quantification ACM AM+13HODE 4.75 ± 0.92 vs. ACM AM+NAC 2.38 ± 0.61; $P < 0.0001$; ORO quantification ACM AM+13HODE 4.75 ± 0.92 vs. ACM AM+13HODE+NAC 3.48 ± 0.78; $P = 0.02$; Fig 4B). Accordingly, both PPARγ and CD36 protein levels were reduced ($n = 7$; PPARγ/GAPDH d.a. ACM AM+13HODE 2.17 ± 0.20 vs. ACM AM+NAC 1.03 ± 0.08; $P < 0.0001$; PPARγ/GAPDH d.a.

ACM AM+13HODE 2.17 ± 0.20 vs. ACM AM+13HODE+NAC 1.31 ± 0.15; $P = 0.0001$; $n = 8$; CD36/GAPDH d.a. ACM AM+13HODE 1.72 ± 0.20 vs. ACM AM+NAC 0.95 ± 0.10; $P < 0.0001$; CD36/GAPDH d.a. ACM AM+13HODE 1.72 ± 0.20 vs. ACM AM+13HODE+NAC 0.86 ± 0.15; $P < 0.0001$; Fig 4C).

Moreover, NAC addition to AM led to lipid accumulation reduction in ACM C-MSC compared to AM alone, suggesting that oxidative stress plays a role in the lipogenic process *per se* ($n = 12$; ORO quantification ACM AM 3.60 ± 0.82 vs. ACM AM+NAC 2.38 ± 0.61; $P = 0.03$; $n = 7$; PPARγ/GAPDH d.a. ACM AM 1.67 ± 0.16 vs. ACM AM+NAC 1.03 ± 0.08; $P = 0.007$; Fig 4B and C). The effect is possibly due to a direct action of NAC on gene expression as previously described in macrophages during foam cell formation (Sung *et al*, 2012).

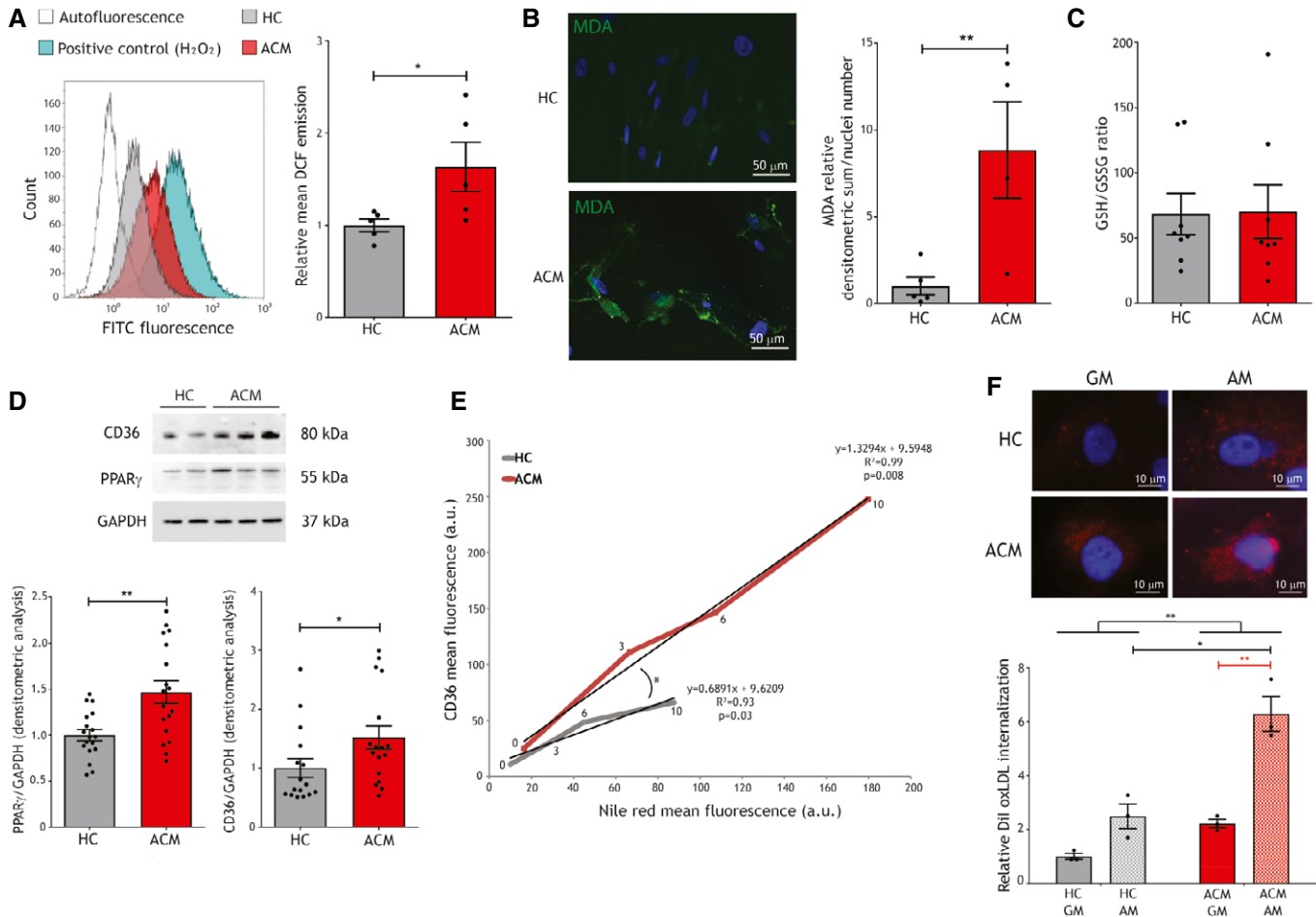

**Figure 3. ACM C-MSC show elevated oxidative stress, PPARγ, and CD36 expression, and, during adipogenic differentiation, CD36 levels and functionality.**

A  Left panel: exemplificative FACS analysis of the FITC emission of unstained HC C-MSC (autofluorescence; white), HC C-MSC in GM treated with DCF (gray), ACM C-MSC in GM treated with DCF (red), and HC cells in GM treated with 2 mmol/l H₂O₂ as positive control (blue). Right panel: mean DCF emission of HC and ACM C-MSC in GM ($n = 5$ biological replicates; two-tailed Student's $t$-test).

B  Left panels: representative images of MDA immunostaining (green) on ACM and HC C-MSC in GM. Nuclei are counterstained with Hoechst33342 (blue). Right panel: image quantification ($n = 4$ biological replicates ACM, $n = 5$ biological replicates HC; two-tailed Student's $t$-test).

C  GSH/GSSG ratio quantification in HC and ACM C-MSC cultured in GM ($n = 8$ biological replicates).

D  Top panel: representative images of Western blot analysis of proteins extracted from ACM and HC C-MSC cultured in GM, hybridized with anti-CD36 and anti-PPARγ antibodies. Immunostaining of the housekeeping GAPDH is shown for normalization. Bottom panel: d.a. of PPARγ ($n = 18$ biological replicates) and CD36 ($n = 17$ vs. $n = 16$ biological replicates) levels, normalized on GAPDH (Mann–Whitney test).

E  Results of a FACS analysis of ACM and HC C-MSC, cultured in GM (time-point 0) or AM for 3, 6 or 10 days (time-points 3, 6,and 10, respectively) and marked with anti-CD36 antibody and Nile red. The mean ($n = 4$ biological replicates ACM, $n = 5$ biological replicates HC) fluorescence of CD36 and Nile Red is shown for each condition, the relative regression line, its equation, $R^2$, and $P$-value (X-Y correlation).

F  Top panels: representative images of internalization of oxLDL (red) in HC and ACM cells, cultured either in GM or in AM, and subjected to 10 μg/ml DiI-oxLDL treatment. Nuclei are counterstained with Hoechst33342 (blue). Bottom panel: quantification of the relative mean DiI fluorescence for each sample, measured by FACS analysis ($n = 3$ biological replicates; two-way ANOVA).

Data information: mean ± SEM. *$P < 0.05$ and **$P < 0.01$.
Source data are available online for this figure.

## CD36 silencing and PPARγ inhibition in ACM C-MSC impair the lipid accumulation mechanism

To directly assess CD36 causal role in ACM cell adipogenic process, we performed CD36 siRNA-mediated silencing in AM supplemented with oxLDL. 32% mean reduction in CD36 levels ($n = 7$; $P = 0.014$; Fig 5A) in ACM C-MSC was enough for inducing significantly less lipid accumulation if compared to the non-silenced counterparts

($n = 7$; ORO relative lipid accumulation scramble $1.00 \pm 0.19$ vs. siRNA $0.35 \pm 0.10$; $P = 0.003$; Fig 5B), along with PPARγ reduced levels ($n = 7$; PPARγ/GAPDH relative d.a. scramble $1.00 \pm 0.18$ vs. siRNA $0.76 \pm 0.15$; $P = 0.05$; Fig 5A). In addition, CD36 levels (both with and without silencing) correlated with PPARγ ($n = 14$; slope=1.05; $R^2 = 0.76$; $P = 0.0002$) and to a lesser extent with ORO staining ($n = 14$; slope=0.52; $R^2 = 0.26$; $P = 0.07$; Fig 5C). As expected, reduced CD36 levels determined lower oxLDL

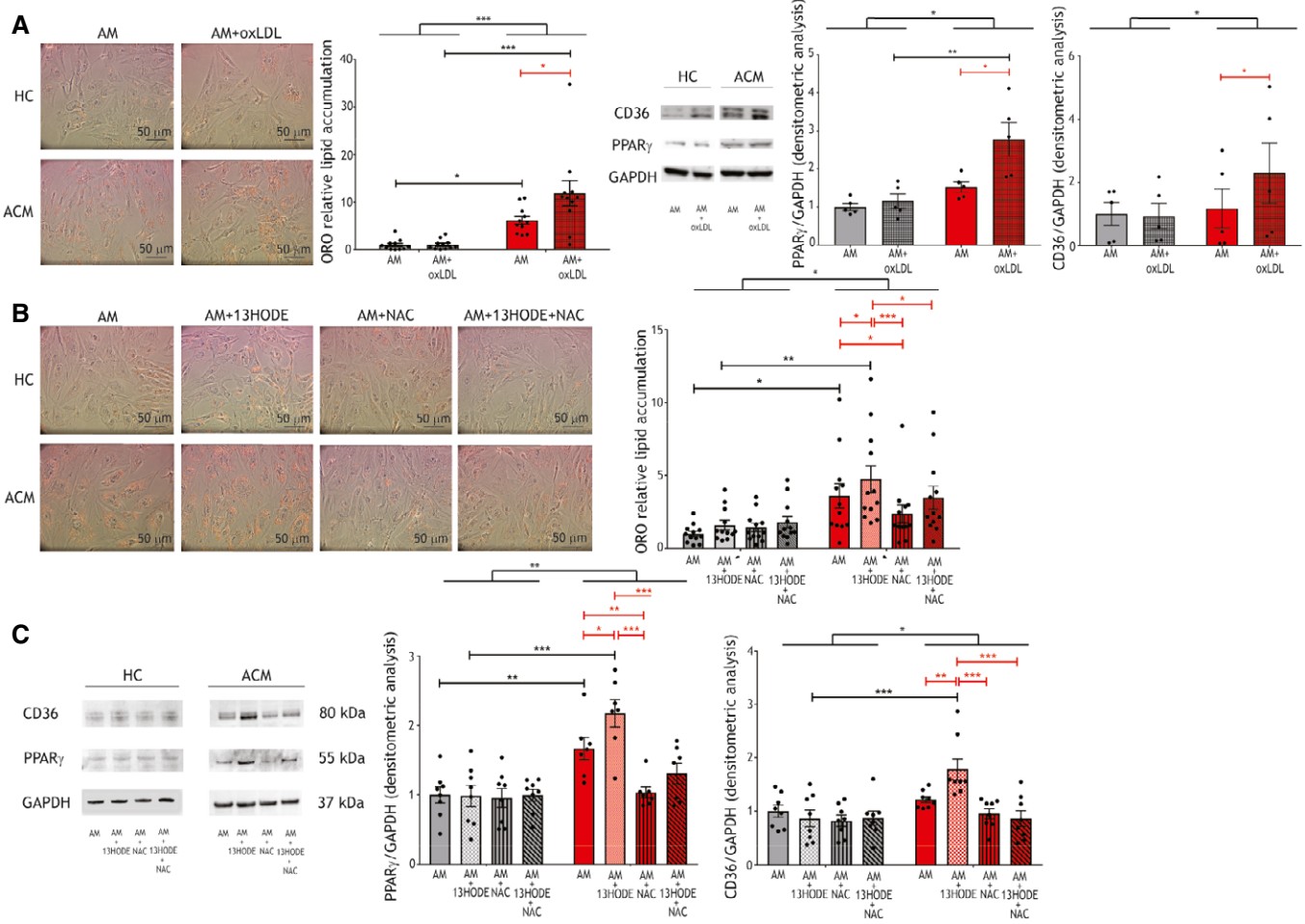

**Figure 4. ACM C-MSC lipogenesis is increased with oxLDL or 13HODE and prevented by NAC treatment.**

A   Left panels: representative images of ORO staining on ACM and HC C-MSC in AM supplemented or not with 150 μg/ml oxLDL. Middle panel: image quantification
    (n = 11 biological replicates; two-way ANOVA). Right panels: representative images of Western blot of CD36, PPARγ, and GAPDH protein expression of ACM and HC C-
    MSC protein extracts in AM supplemented or not with 150 μg/ml oxLDL (n = 5 biological replicates) and d.a. normalized on the housekeeping GAPDH (two-way
    ANOVA).
B   Left panels: representative images of ORO staining on ACM and HC C-MSC in AM supplemented or not with 20 μg/ml 13HODE, 5 mmol/l NAC, or both. Right panel:
    image quantification (n = 13 biological replicates; two-way ANOVA).
C   Left panel: representative images of Western blot of CD36, PPARγ, and GAPDH protein expression of ACM and HC C-MSC protein extracts in AM supplemented or not
    with 20 μg/ml 13HODE, 5 mmol/l NAC, or both (n = 8 biological replicates). Right panels: d.a. normalized on the housekeeping GAPDH (two-way ANOVA).

Data information: mean ± SEM. *P < 0.05, **P < 0.01, and ***P < 0.001.
Source data are available online for this figure.

internalization (n = 3; DiI internalization scramble 7.34 ± 1.31 vs. siRNA 0.63 ± 0.13; P = 0.04; Fig 5D).

oxLDL/CD36/PPARγ interdependence in ACM cells was also confirmed by inhibiting PPARγ with the antagonist GW9662. This provoked a significant reduction of oxLDL internalization (n = 3; DiI internalization ACM AM 25,137 ± 3,567 vs. ACM AM+GW9662 10,194 ± 3,787; P = 0.01; Fig 5E), together with a lower expression of CD36 (n = 3; CD36/GAPDH d.a. ACM AM 1.26 ± 0.08 vs. ACM AM+GW9662 0.97 ± 0.14; P = 0.04; Fig 5F).

In addition, we confirmed that lipid accumulation and CD36 levels and functionality increased also in human ACM cardiomyocytes (using hiPSC-CM derived from ACM patients vs. unaffected relatives) upon PPARγ agonism (Appendix Fig S5).

**HFD administration leads to cardiac lipid accumulation and dysfunction in *Pkp2*+/− mice**

To validate our hypothesis on an ACM *in vivo* model, we took advantage of the well-established *Pkp2*+/− mouse (Grossmann *et al*, 2004; Sato *et al*, 2009). Despite adult *Pkp2*+/− mice showed no cardiac adipogenesis, low oxidative stress and CD36 levels, and no cardiac dysfunction, C-MSC obtained from the mutant mice accumulated lipids *in vitro* (Appendix Figs S6 and S7).

We thus fed *Pkp2*+/− mice a HFD for 3 months to test the hypothesis that increasing cholesterol and oxidative stress levels (Matsuzawa-Nagata *et al*, 2008) would promote cardiac lipid accumulation.

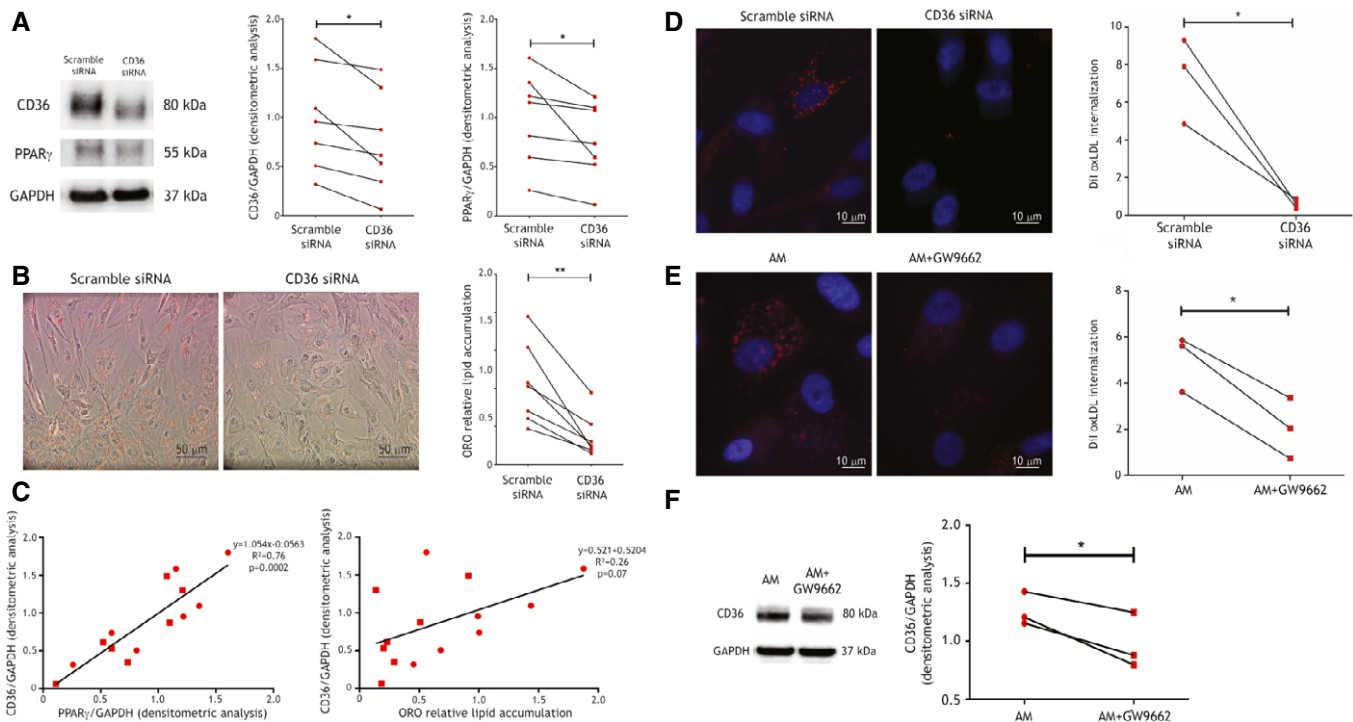

**Figure 5. CD36 silencing reduces lipid accumulation and PPARγ antagonism reduces CD36 expression and function in ACM C-MSC.**

A   Left panel: representative images of Western blot of CD36, PPARγ, and GAPDH expression of protein extracts of ACM C-MSC treated with scramble siRNA or CD36 siRNA. Right panels: d.a. normalized on the housekeeping protein GAPDH (n = 7 biological replicates; two-tailed Student's t-test).

B   Left panels: representative images of ORO staining on ACM C-MSC treated with scramble siRNA or CD36 siRNA and cultured in AM with 150 μg/ml oxLDL. Right panel: image quantification (n = 7 biological replicates; two-tailed Student's t-test).

C   Left panel: correlation between CD36/GAPDH and PPARγ/GAPDH d.a. Right panel: correlation between CD36/GAPDH d.a. and ORO lipid accumulation. Regression lines, their equations, R², and correlation P values are shown (X-Y correlation).

D   Left panel: representative images of internalization of oxLDL (red) in ACM C-MSC treated with scramble siRNA or CD36 siRNA, cultured in AM and subjected to 10 μg/ml DiI-oxLDL treatment. Right panel: quantification of the DiI fluorescence normalized on nuclei number (n = 3 biological replicates, arbitrary units; two-tailed Student's t-test).

E   Left panel: representative images of internalization of oxLDL (red) in ACM C-MSC cultured in AM or AM+5 μM GW9662 and subjected to 10 μg/ml DiI-oxLDL treatment. Nuclei are counterstained with Hoechst33342 (blue). Right panel: quantification of the DiI fluorescence normalized on nuclei number for each sample (n = 3 biological replicates, arbitrary units; two-tailed Student's t-test).

F   Left panel: representative images of Western blot of CD36 and GAPDH expression of protein extracts of ACM C-MSC cultured in AM or AM+5 μM GW9662. Right panels: d.a. normalized on the housekeeping GAPDH (n = 7 biological replicates; two-tailed Student's t-test).

Data information: mean ± SEM. *P < 0.05 and **P < 0.01.
Source data are available online for this figure.

In both strains, plasma total cholesterol concentrations in the lipoprotein fractions comparably increased after HFD. In particular, higher cholesterol was measured in HFD in fractions 29–32, representing VLDL-IDL-LDL (n = 8; P < 0.0001 each fraction; Fig 6A). Of note, HFD also increased oxLDL (n = 6; oxLDL nmol/ml HFD 1.29 ± 0.11 vs. CD 0.83 ± 0.05; P = 0.049; Fig 6B).

Following HFD, cardiac sections from Pkp2+/− mice showed larger areas of fatty substitution compared to WT siblings (n = 10; % ORO positive area Pkp2+/− 1.04 ± 0.24 vs. WT 0.13 ± 0.02; P < 0.0001; Fig 6C), prevalently in sub-epicardial areas. As expected, the cells undergoing adipogenic differentiation in murine hearts are of mesenchymal origin (Appendix Fig S8). Accordingly, the difference in PPARγ immunoreactivity between WT and Pkp2+/− hearts substantially increased with HFD (n = 10; PPARγ d.a./nuclei number Pkp2+/− 8.66 ± 1.59 vs. WT 2.13 ± 0.79; P = 0.0001; Fig 6D; Appendix Fig S9).

Pkp2+/− mouse hearts also showed oxidative stress (n = 10; MDA relative d.a./nuclei number Pkp2+/− 10.43 ± 2.19 vs. WT 1.71 ± 1.21; P = 0.0003; Fig 6E; Appendix Fig S8) and CD36 level increase (n = 10; CD36 relative d.a./nuclei number Pkp2+/− 7.11 ± 1.72 vs. WT 2.99 ± 0.68; P = 0.04; Fig 6F; Appendix Fig S8). Moreover, HFD induced RV dysfunction but no changes in LV function, as assessed by 2D echocardiography. In particular, ACM mice after 3 months of diet presented a lower RV ejection fraction (EF), fractional shortening (FS), and RV internal diameter (RVID) in systole compared to WT (n = 10; %RV EF Pkp2+/− 73.44 ± 3.17 vs. WT 82.66 ± 1.36; P = 0.02; %RV FS Pkp2+/− 40.11 ± 2.74 vs. WT 48.14 ± 1.55; P = 0.02; RVID systole Pkp2+/− 0.93 ± 0.05 vs. WT 0.75 ± 0.05 mm; P = 0.03; Fig 6G). Morphometric analyses are shown in Appendix Fig S10. Accordingly, mouse electric activity resulted impaired by the combination of the Pkp2+/− genotype and HFD, as shown by electrocardiogram

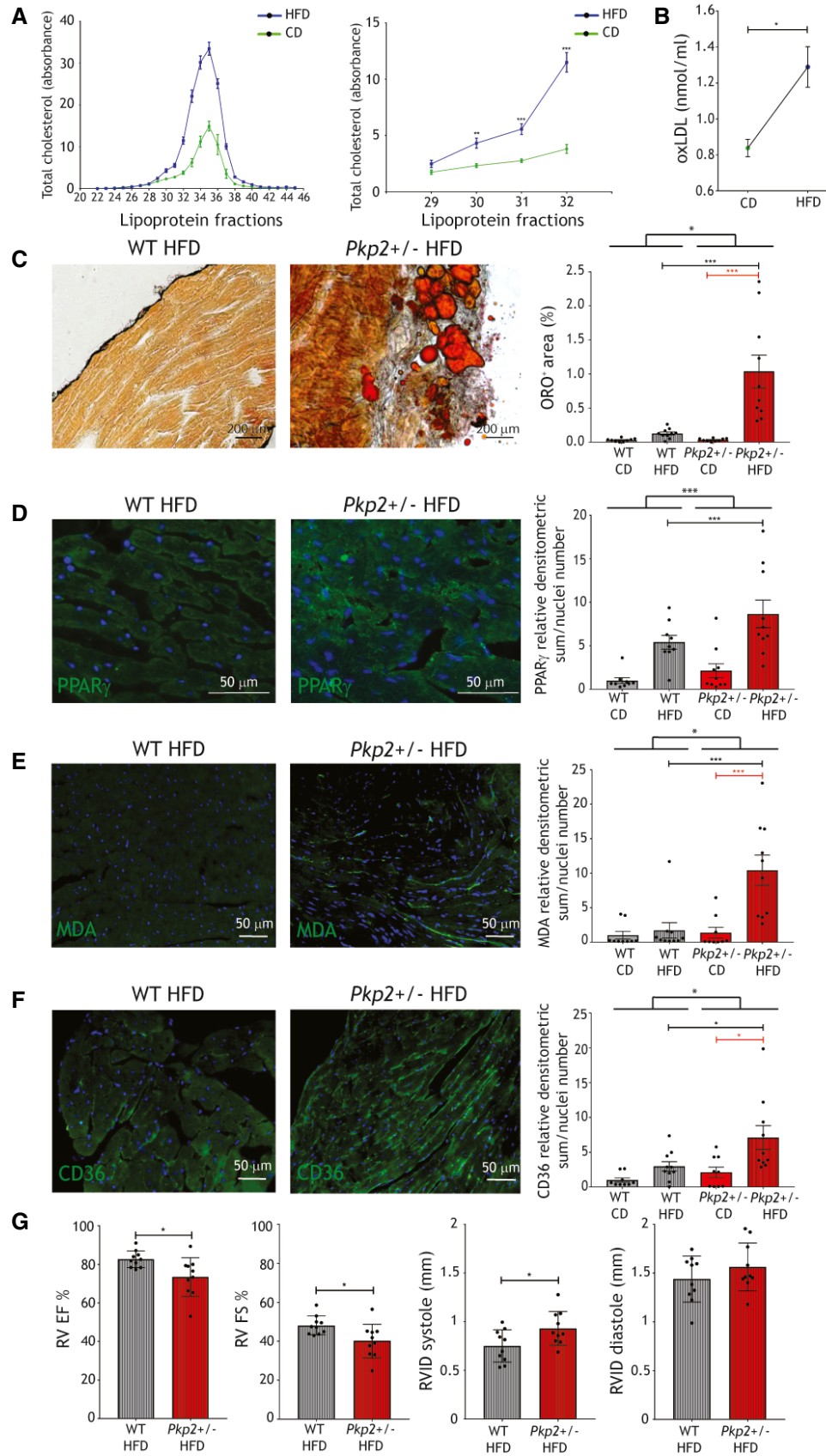

**Figure 6.**

**Figure 6.   HFD increases plasma oxLDL, cardiac adipose substitution, PPARγ and CD36 expression, and lipid peroxidation, and impairs RV function in ACM mice.**

A   Left panel: plasma total cholesterol in the different lipoprotein fractions of WT and *Pkp2*+/− mice fed a CD (green line) or a 3-month HFD (blue line; $n = 7$). Right panel: plasma total cholesterol quantity in fractions 29–32, corresponding to low-, very-low-, and intermediate-density lipoproteins, of WT and *Pkp2*+/− mice fed a CD (green line) or a 3-month HFD (blue line; $n = 7$; Mann–Whitney test).

B   Plasma concentration of oxLDL in WT and *Pkp2*+/− mice samples, fed a CD and a 3-month HFD ($n = 6$; two-tailed Student's *t*-test).

C   Left panels: representative images of ORO staining of HFD-fed WT and *Pkp2*+/− cardiac sections. Right panel: quantification of ORO positive area percentage ($n = 10$). For comparison, quantification of ORO positive area of cardiac sections of CD-fed WT and *Pkp2*+/− mice ($n = 9$; two-way ANOVA) is shown (Appendix Fig S3).

D   Representative images of PPARγ (green) immunostaining on HFD-fed WT and *Pkp2*+/− mice cardiac sections ($n = 10$; two-way ANOVA). Quantification of the PPARγ staining in HFD is shown relative to the values of CD ($n = 9$).

E   Representative images of MDA (green) immunostaining on HFD-fed WT and *Pkp2*+/− mice cardiac sections ($n = 10$; two-way ANOVA). Quantification of the MDA staining in HFD is shown relative to the values of CD ($n = 9$).

F   Representative images of CD36 immunostaining (green) on HFD-fed WT and *Pkp2*+/− mice cardiac sections ($n = 10$; two-way ANOVA). Nuclei are counterstained with Hoechst33342 (blue). Quantification of the staining in HFD is shown relative to the values of CD ($n = 9$).

G   RV EF and FS percentages, RVID in systole and diastole of WT and *Pkp2*+/− mice after (right) 3-month HFD feeding ($n = 10$; two-tailed Student's *t*-test).

Data information: mean ± SEM. *$P < 0.05$; **$P < 0.01$; ***$P < 0.001$.
Source data are available online for this figure.

(ECG) trace change and CX43 reduction and mislocalization (Appendix Fig S11).

### Atorvastatin administration prevents ACM manifestation in HFD-fed *Pkp2*+/− mice

To evaluate the effect of the pharmacological counteraction of high oxLDL levels (Tsimikas *et al*, 2004), we fed WT and *Pkp2*+/− mice a 3-month HFD supplemented with 20 mg/kg atorvastatin. In both strains, lower cholesterol was measured in HFD+atorvastatin in fractions 29–32, representing VLDL-IDL-LDL ($n = 8$; $P < 0.001$; Fig 7A). Moreover, in parallel with plasma oxLDL concentration reduction ($n = 6$ vs. $n = 8$; oxLDL nmol/ml HFD $1.29 \pm 0.11$ vs. HFD+atorva $0.56 \pm 0.05$; $P = 0.009$; Fig 7B), we observed lower cardiac lipid accumulation ($n = 10$ vs. $n = 9$; %ORO+ area *Pkp2*+/− HFD $1.04 \pm 0.24$ vs. *Pkp2*+/− HFD+atorva $0.02 \pm 0.001$; $P = 0.001$; Fig 7C) and a dramatic decrease of PPARγ expression levels ($n = 10$ vs. $n = 9$; PPARγ d.a./nuclei number *Pkp2*+/− HFD $8.65 \pm 1.59$ vs. *Pkp2*+/− HFD+atorva $0.02 \pm 0.006$; $P < 0.0001$; Fig 7D; Appendix Fig S9). MDA expression also significantly decreased ($n = 10$ vs. $n = 9$; MDA relative d.a./nuclei number *Pkp2*+/− HFD $10.43 \pm 2.19$ vs. *Pkp2*+/− HFD+atorva $0.02 \pm 0.007$; $P = 0.0003$; Fig 7E; Appendix Fig S9) as well as CD36 levels ($n = 10$ vs. $n = 9$; CD36 relative d.a./nuclei number *Pkp2*+/− HFD $7.11 \pm 1.72$ vs. *Pkp2*+/− HFD+atorva $0.03 \pm 0.01$; $P = 0.001$; Fig 7F; Appendix Fig S9). Moreover, RV EF, RV FS, and RVID in systole improved in *Pkp2*+/− HFD+atorva with respect to HFD only ($n = 10$ vs. $n = 9$; %RV EF *Pkp2*+/− HFD $73.44 \pm 3.17$ vs. *Pkp2*+/− HFD+atorva $82.14 \pm 1.74$; $P = 0.04$; %RV FS *Pkp2*+/− HFD $40.11 \pm 2.74$ vs. *Pkp2*+/− HFD+atorva $47.94 \pm 2.01$; $P = 0.04$; Fig 7G). Defects in the ECG were prevented by atorvastatin treatment, as well as those in CX43 expression and localization (Appendix Fig S12).

## Discussion

Current information about ACM genetic basis does not fully explain reduced penetrance and phenotypic variability (Pinamonti *et al*, 2014). Among ACM pathogenic pathways, metabolic dysfunctions are thought to play a relevant role, given the well-established involvement of PPARγ and recent discoveries on the topic (van Opbergen *et al*, 2019; Song *et al*, 2020).

In the present study, we defined the contribution of oxLDL/CD36/PPARγ circuitry as ACM penetrance cofactor, demonstrating, for the first time, that oxidative stress and oxidized lipid metabolism modulate ACM adipogenic phenotype, both *in vitro* and *in vivo*. Interestingly, such biological axis is pharmacologically targetable in order to reduce the adipogenic phenotype and consequent disease severity in ACM.

This hypothesis-driven study was prompted by the observations in ACM patient plasma of higher oxLDL and 13HODE plasma levels as compared to matched HC. Our cohort was composed of 81% male and 38% athletic patients. Of note, strenuous exercise and male sex are the only accepted cofactors precipitating ACM phenotype. The association of both exercise and male gender with oxidative stress is well known (Miller *et al*, 2007). Oxidative stress can induce oxidation of different proteins and complexes, including LDL (Sanchez-Quesada *et al*, 1995).

Our data suggest that a higher oxLDL concentration in ACM patient plasma is not a mere consequence of the causative genetic defect. Indeed, we observed that oxLDL plasma levels are higher in ACM patients than in their unaffected relatives, carriers of the same ACM causative genetic mutation. Further, oxLDL elevation is not linked to a specific genetic form of ACM, even if, given the low prevalence in our cohort of carriers of pathogenic mutations in desmosomal genes other than *PKP2*, testing each single gene association was not possible. On the other hand, we identified background variants in genes associated with oxidative stress or dyslipidemia co-segregating with the ACM phenotype (Appendix Fig S1; Appendix Table S4; Dataset EV1). These results will need confirmation in larger cohorts. Other factors, including lifestyle and diet, or unidentified protective genetic variants, may play a role in modulating oxLDL concentrations.

Notably, we unraveled in our ACM patients a strong association between oxLDL plasma levels above the cut-off of 86 ng/ml and pathognomonic ACM structural and clinical features. Such circulating oxLDL cut-off allowed to segregate ACM patient population with a severe clinical phenotype in terms of fat infiltration, ventricular dysfunction, and risk of major arrhythmic events in the long term. In addition, these data are relevant to confirm oxLDL pathogenic role at a clinical level and to introduce oxLDL plasma levels as new

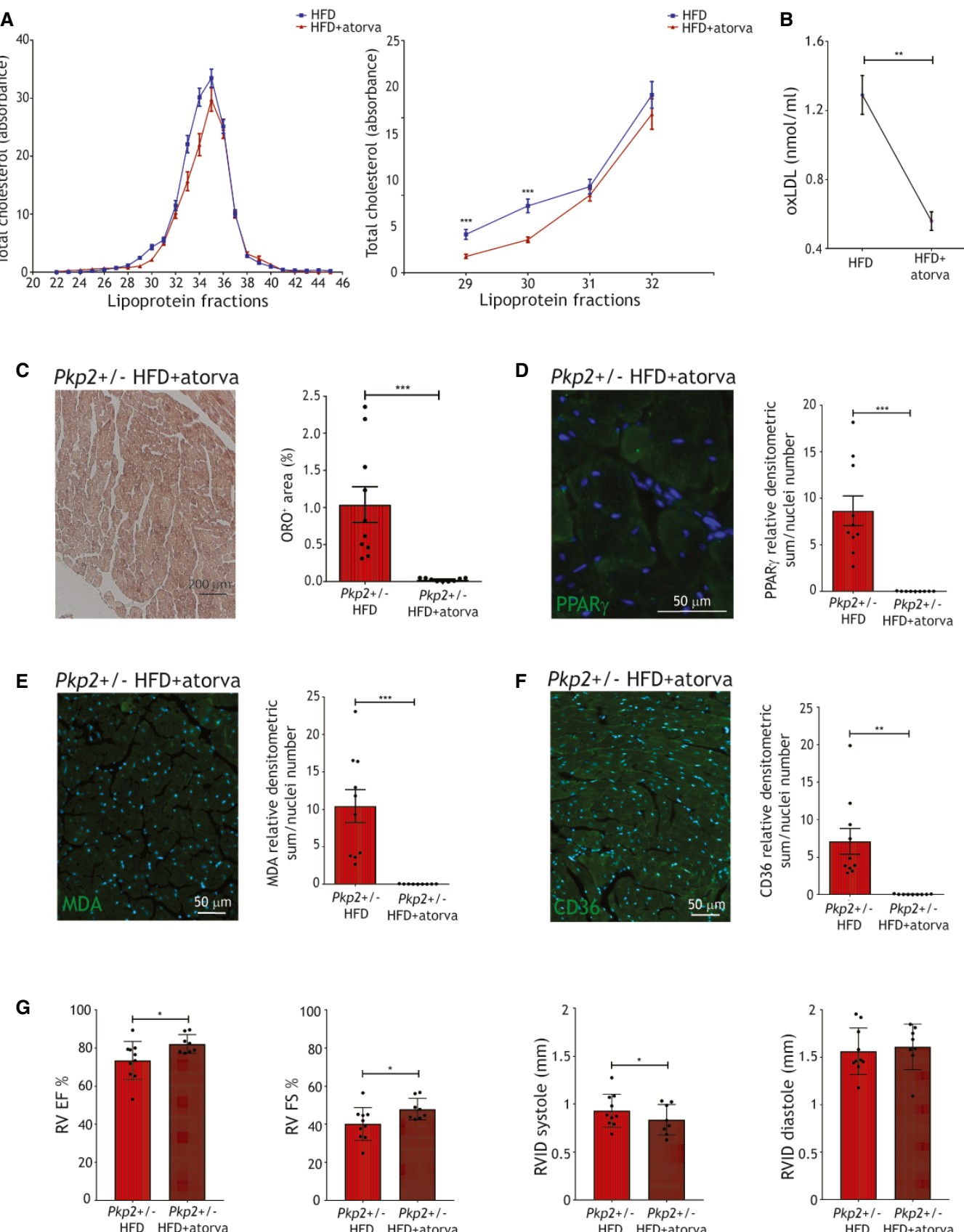

**Figure 7.**

**Figure 7. Atorvastatin prevented ACM phenotype in HFD-fed *Pkp2*+/− mice.**

A   Left panel: plasma total cholesterol in the different lipoprotein fractions of WT and *Pkp2*+/− mice fed a 3-month HFD (blue line) or a 3-month HFD+atorvastatin (red line). Right panel: plasma total cholesterol quantity in fractions 29–32, corresponding to low-, very-low-, and intermediate-density lipoproteins, of WT and *Pkp2*+/− mice fed a 3-month HFD (blue line; *n* = 7) or a 3-month HFD+atorvastatin (red line; *n* = 9; two-tailed Student's *t*-test).

B   oxLDL plasma concentration in WT and *Pkp2*+/− mouse samples, fed a 3-month HFD plus atorvastatin (*n* = 8), compared to HFD (as in Fig 6B; two-tailed Student's *t*-test).

C   Left panel: representative images of ORO staining of cardiac sections of *Pkp2*+/− mice, fed a 3-month HFD plus atorvastatin. Right panel: quantification of the percentage of ORO positive area (*n* = 9) is compared to that in HFD (as in Fig 6C; two-tailed Student's *t*-test).

D   Representative images of PPARγ immunostaining (green) on cardiac sections of *Pkp2*+/− mice fed a 3-month HFD plus atorvastatin (*n* = 9). Quantification is compared to the values of *Pkp2*+/− in HFD and relative to WT in CD (as in Fig 6D; two-tailed Student's *t*-test).

E   Representative images of MDA immunostaining (green) on cardiac sections of *Pkp2*+/− mice fed a 3-month HFD plus atorvastatin (*n* = 9). Quantification is compared to the values of *Pkp2*+/− in HFD and relative to WT in CD (as in Fig 6E; two-tailed Student's *t*-test).

F   Representative images of CD36 immunostaining (green) on cardiac sections of *Pkp2*+/− mice fed a 3-month HFD plus atorvastatin (*n* = 9). Quantification is compared to the values of *Pkp2*+/− HFD and relative to WT in CD (as in Fig 6F; two-tailed Student's *t*-test). Nuclei are counterstained with Hoechst33342 (blue).

G   RV EF and FS percentages, RVID in systole and diastole of *Pkp2*+/− mice during (left panel) and after (right panel) 3-month HFD plus atorvastatin feeding (*n* = 9). For comparison, RV EF of *Pkp2*+/− mice fed a HFD is shown (as in Fig 6G; two-tailed Student's *t*-test).

Data information: mean ± SEM. *$P < 0.05$; **$P < 0.01$; and ***$P < 0.001$.
Source data are available online for this figure.

potential circulating prognostic marker in ACM (Stadiotti *et al*, 2018).

We used human C-MSC obtained from ACM heart biopsies and HC as *in vitro* model, as they subsidize ACM-driven cardiac adipogenesis and represent a reliable cell model to study pathogenic mechanisms (Sommariva *et al*, 2016; Sommariva *et al*, 2017). As previously demonstrated in other cell models (Garcia-Gras *et al*, 2006; Chen *et al*, 2014; Sommariva *et al*, 2017), in ACM C-MSC we found a higher expression of PPARγ. In agreement with the oxLDL receptor CD36 interdependence with PPARγ, described in cardiovascular atherogenic conditions (Nicholson *et al*, 1995), we found also in ACM C-MSC a linear correlation between the activation of PPARγ, with consequent lipid accumulation, and CD36 expression and function. The proposed mechanism was confirmed by CD36 silencing, which entailed a significant reduction of lipid accumulation in ACM C-MSC, as well as by PPARγ inhibition, which caused a reduction of CD36 expression and function. Moreover, we described, for the first time, that increased oxidative stress is present in ACM C-MSC, without changes in their antioxidant capability. Thus, an analysis of mitochondrial dysfunction as a source of oxidative stress is awaited (van Opbergen *et al*, 2019).

We further demonstrated that oxLDL enhances C-MSC adipogenic differentiation. Accordingly, Parhami *et al* (1999) have previously shown that oxLDL promotes adipogenic differentiation of bone marrow-derived MSC. We showed that 13HODE prompts, in ACM patient C-MSC, a PPARγ/CD36-dependent adipogenesis as an intrinsic ACM pathogenic mechanism. 13HODE effect was previously used in ACM hiPSC-CM as a mere *in vitro* supplement to activate PPARγ (Kim *et al*, 2013). Our results do not exclude that the Wnt/βcatenin pathway may exert a modulatory role in the balance of PPARγ activation. In addition, we showed that PPARγ-agonism induces CD36 increased expression and function in ACM hiPSC-CM. Importantly, cell exposure to the antioxidant NAC not only prevented 13HODE-dependent lipogenesis and CD36/PPARγ expression, but ameliorated C-MSC phenotype, possibly by reducing the effects of basal oxidized lipids in their ability to activate PPARγ, as well as directly acting on the transcriptional profile of the cells, including CD36 and PPARγ expression (Ji *et al*, 2011; Sung *et al*, 2012).

We confirmed *in vitro* findings with *in vivo* experiments by means of the *Pkp2*+/− mouse model (Grossmann *et al*, 2004), since *PKP2* heterozygous mutations are the most frequent in ACM patients. Notably, available ACM *in vivo* models, including the *Pkp2*+/− mouse, do not fully recapitulate the disease phenotype, showing from absence to small amount of ventricular fibro-fatty substitution, possibly due to an intrinsic protective mechanism. Since it is known that mice show low cholesterol plasma levels, we demonstrated that a consistent increase of plasma LDL cholesterol, including the oxidized form, achieved by HFD feeding, provoked fat accumulation in ACM mouse hearts. This phenomenon, borne by C-MSC, as previously proven in humans (Sommariva *et al*, 2016), was prevalently located in sub-epicardial regions, in line with the epicardial-endocardial gradient observed in ACM human hearts. In accordance with *in vitro* and patient *ex vivo* results, cardiac oxidative stress and CD36 significantly increased in ACM mouse hearts following HFD. Importantly, HFD treatment provoked an initial impairment of RV function, in line with the concept that myocardial substrate alterations lead to functional impairment (Pabon *et al*, 2018; te Riele *et al*, 2014) and adipocyte-secreted factors negatively influence CM contractility (Lamounier-Zepter *et al*, 2009; Gastaldelli *et al*, 2012; Lamounier-Zepter *et al*, 2014; Pabon *et al*, 2018). In addition, proarrhythmic (Peters *et al*, 2012; Canpolat *et al*, 2013) ECG changes were detected in the *Pkp2*+/− mice fed HFD. These results confirmed that ACM causative mutations are necessary but not sufficient to generate an overt ACM phenotype in mice, which was instead obtained when increasing oxLDL levels, thus recognizing in oxLDL a likely contributory pathogenic cause.

We cannot exclude that disease progression itself may, in turn, further increase oxLDL levels in patients, e.g., by boosting ROS production, inflammation, or other mechanisms. However, our *in vitro*, *in vivo,* and clinical data clearly showed that maintaining the levels of oxLDL low is beneficial in terms of disease progression.

Remarkably, atorvastatin administration prevented cardiac fat accumulation and RV dysfunction in HFD-fed *Pkp2*+/− mice. Beyond the described canonical lipid-lowering effects, we infer that known pleiotropic effects, as anti-inflammatory (Antonopoulos *et al*, 2012), DKK- and Wnt pathway- mediated (Pontremoli *et al*, 2018), as well as sympathetic activity reducing (Lewandowski *et al*, 2015), and antiarrhythmic (Kostapanos *et al*, 2007), may contribute to ameliorate ACM cardiac phenotype.

An interesting side-product of this project was the generation of a novel diet-driven ACM mouse model showing fat accumulation and RV dysfunction, thus better recapitulating, beyond arrhythmias (Cerrone *et al*, 2012), human ACM substrate myocardial phenotype.

In conclusion, by means of a multilayer approach, we demonstrated, for the first time, that oxidative stress increases oxLDL bioavailability, which are internalized in C-MSC by CD36 receptor, thus acting, through 13HODE-mediated PPARγ activation, as a cofactor of cardiac adipogenic differentiation and with dependent phenotypes (Synopsis figure), which can be targeted by available therapies such as antioxidants (Shafiei *et al*, 2018) or statins (Ky *et al*, 2008). According to our findings, these strategies have the potential to reduce the penetrance of the disease in ACM mutation carriers, by attenuating cardiac adipose substitution, ventricular dysfunction, and arrhythmic phenotypes.

# Materials and Methods

## Ethics statement

This study complies with the WMA Declaration of Helsinki and the Department of Health and Human Services Belmont Report. It was approved by "IEO-CCM IRCCS" (12/06/2012) and by "South Tyrol Azienda Sanitaria" (13/03/2014, No. 1/2014) Ethics Committees. Written informed consent was obtained from all participants. HC cardiac samples were obtained from donors (accidental death), from "Treviso Tissue Bank Foundation".

## Study patient population

A total of 67 ACM patients were enrolled for this study. ACM diagnosis was reached according to the 2010 International Task Force criteria (Marcus *et al*, 2010). Thirty-six patients out of the total cohort were matched for age, sex, and cardiovascular risk factors to 36 HC without a previous history of heart disease for plasma analysis (Appendix Table S1 summarizes baseline characteristics of 36 ACM patients and 36 HC, and Appendix Table S3 describes the genetic profile of ACM patients). We further enrolled for the plasma analysis nine ACM patients' relatives with *PKP2* mutations but no clinical signs of the disease.

We obtained blood samples from all the recruited ACM patients and HC. ACM patients and HC taking statins or other lipid-lowering drugs were excluded from the analysis. Clinical data were collected for all ACM patients, as available: RV EF % was determined by MRI; RV dysfunction was defined as in Li and Durbin (2010); and MAEs are defined as sustained ventricular tachycardia, ventricular fibrillation, appropriate implantable cardioverter device intervention, and aborted sudden cardiac death. Genetic data on ACM-linked genes were obtained for 65 out of 67 patients (Appendix Table S5).

RV samples were obtained by biopsy procedures from 19 ACM patients.

## Plasma preparation

Blood samples (5 ml) were collected in EDTA-coated tubes and centrifuged at 1,500 *g* for 15 min. Supernatants were collected,

centrifuged again at 16,000 *g* for 15 min to obtain cell- and platelet-free plasma, and stored at −80°C as 400 μl aliquots until usage.

## oxLDL in human plasma samples

Plasma samples from ACM patients and HC were used for the determination of oxLDL quantity, using a specific ELISA kit (Immundiagnostik, Bensheim, Germany). The test recognizes MDA-modified apolipoprotein B100. The protocol recommended by manufacturers was observed, and the absorption was determined with a spectrophotometer (Berthold Technologies, Bad Wildbad, Germany) at 450 nm. Results were interpreted by constructing a dose/response curve according to the standards provided in the kit.

## Genetic analysis

### ACM-associated genes
DNA was extracted from blood or C-MSC using QIAamp DNA Mini kit (Qiagen, Hilden, Germany). Next-generation sequencing was performed with the TruSight™ Cardio Sequencing Kit (Illumina NextSeq, San Diego, CA, USA). The alignment of sequence reads to reference human genome (GRCh37/hg19) was performed using GATK software. Variants in *DSC2*, *DSG2*, *DSP*, *PKP2*, *JUP*, *TMEM43*, *DES*, *RYR2*, *PLN*, *SCN5A*, and *LMNA* were filtered with Wannovar and pathogenicity classified according to Richards *et al* (2015).

### Exome sequencing
NA samples from blood were prepared following the Nextera® Rapid Capture Exome Enrichment kit protocol. Libraries for all samples, except Fam3.I.3, Fam3.I.4, and Fam3.II.3 (Appendix Fig S1), were sequenced on two lanes of an Illumina HiSeq in paired-end mode and a read length of 100 bp. The remaining three samples were sequenced subsequently on four lanes of an Illumina MiSeq in paired-end mode and a read length of 75 bp. Nextera adapters were trimmed off using SeqPrep (https://github.com/jstjohn/SeqPrep), and read quality was controlled with FastQC (http://www.bioinformatics.babraham.ac.uk/projects/fastqc/). Reads were aligned to reference genome GRCh37 with BWA version 0.7.15 (DePristo *et al*, 2011). Duplicate marking was performed with picard tools version 2.8.1 (https://broadinstitute.github.io/picard/). Indels were realigned and base quality scores recalibrated using GATK 3.7 (Van der Auwera *et al*, 2013), following the GATK best practice guidelines (Van der Auwera *et al*, 2013). Quality of bam files was evaluated with QualiMap version 2.2.1 (Garcia-Alcalde *et al*, 2012), sample contamination estimated with verifyBamId (Jun *et al*, 2012), and sex validated by inspection of the X chromosome coverage. Intermediate per sample gvcfs were generated with the GATK HaplotypeCaller, followed by joint genotyping on all individuals with GATK GenotypeGVCFs and additional samples not related to this study. Variants were called on the exonic target regions as defined by the Nextera protocol with a padding of 100 bp around the exons. Variants were annotated with Ensembl gene and variant consequence data using the Dintor gcoords2cons tool (Weichenberger *et al*, 2015). For each variant, the annotation of the transcript with the worst consequence type as according to Ensembl was selected.

First, using the Dintor MendelianFiltration tool, variants were selected that were present either in homo- or heterozygous form in

all affected individuals of all three families, requiring a coverage of at least 10X at variant sites in the respective individuals. Variants were restricted to those mapped to a gene of the oxidative stress and dyslipidemia gene panel (Dataset EV1). Further, variants were removed, if their ExAC EUR allele frequency (AF) (Lek *et al*, 2016), their gnomAD NEF AF (Lek *et al*, 2016), or their 1,000 Genomes phase 3 EUR AF (Auton *et al*, 2015) was greater than 0.3. Variants were not required to be in protein-coding genes (Appendix Table S4A).

Next, each family was analyzed individually. Using the Dintor MendelianFiltration tool, variants were selected that were present in a families' affected *PKP2* carriers, but absent in the same families' healthy *PKP2* carriers, requiring a coverage of at least 10X at the variant sites in the relevant individuals. For the selection, variants segregating in the affected *PKP2* carriers in either a dominant or a recessive mode of inheritance were accepted. Variants were restricted to those mapped to a gene of the oxidative stress and dyslipidemia gene panel (Dataset EV1). Further, variants were removed, if their ExAC EUR allele frequency (AF) (Lek *et al*, 2016), their gnomAD NEF AF (Lek *et al*, 2016), or their 1,000 Genomes phase 3 EUR AF (Auton *et al*, 2015) was greater than 0.05. Variants were not required to be in protein-coding genes (Appendix Table S4B).

Per sample, an average of 55 million±17 million reads mapped to reference genome, resulting in a mean coverage of 57X±21X, and 87%±4% of the exon target region covered at ≥ 10X. The sex of all samples was confirmed by inspection of the coverage of the X chromosome, and sample contamination was below 1%.

## 13HODE quantification

The determination of plasma 13HODE levels was performed by a liquid chromatography–tandem mass spectrometry (LC-MS/MS) method. Briefly, plasma samples (100 µl) were acidified with 1% formic acid, mixed with the internal standard ($d_4$-13HODE, final concentration 50 pg/µl; Cayman Chemicals, Ann Arbor, Michigan, USA), and purified through HLB extraction cartridges (Oasis® HLB 1cc (30 mg), Waters, Milford, Massachusetts, USA). The eluted fraction was evaporated to dryness and reconstituted with 250 µl of water/methanol/acetonitrile (80:10:10, v/v/v) before the LC-MS/MS analysis. The LC-MS/MS analysis was performed using an Accela HPLC System (Thermo Fisher Scientific, Waltham, Massachusetts, USA) coupled to a triple quadrupole mass spectrometer TSQ Quantum Access (Thermo Fisher Scientific, Waltham, Massachusetts, USA) outfitted with electrospray ionization source operating in negative mode. The chromatographic separation was achieved using an XBridge® C18 column (2.1 mm×30 mm, particle size 2.5 µm, Waters) at 30°C. The mobile phase was composed by 2 mM ammonium acetate in water/acetonitrile/methanol (87:10:3, v/v/v, at pH8 by ammonium hydroxide) (solvent A) and 2 mM ammonium acetate in acetonitrile/water/methanol (87:10:3, v/v/v, at pH8 by ammonium hydroxide) (solvent B). The following gradient, at a flow rate of 250 µl/min, was used: 0 min—10% B, 2 min —35% B, 2.5 min—90% B, 6 min—90% B, 6.5 min—10% B, and 12 min—10% B. The analytes were detected by multiple reaction monitoring, and the transitions monitored (precursor ion > product-fragment ions) were $m/z$ 295.1 → $m/z$ 276.8, 194.9 (13HODE) and $m/z$ 299.1 → $m/z$ 279.9, 197.8 ($d_4$-13HODE). A

linear 6-point calibration curve (range 2–62.5 pg/µl) was used for the quantification.

## Biopsy sampling

Endomyocardial biopsy sampling was performed in ACM-suspected patients for diagnostic purposes, guided by CARTO mapping, as previously described (Casella *et al*, 2015; Pilato *et al*, 2018). A biopsy sample from the RV acquired in the area adjacent to the electroanatomical scar was obtained from ACM patients and was processed to obtain heart sections, total tissue protein extracts, and C-MSC (Sommariva *et al*, 2016). HC RV autoptic samples were treated with the same protocols.

## Heart tissue section preparation and immunofluorescence analysis

Human ventricular samples were fixed in 4% paraformaldehyde (Santa-Cruz, Dallas, Texas, USA) in phosphate-buffered saline (PBS; Lonza, Basel, Switzerland) and processed for paraffin embedding. Paraffin-embedded sections (6 µm thick) were de-waxed in xylene and rehydrated in ascending alcohols. The immunofluorescence analysis was performed following antigen retrieval with incubation with target retrieval solution citrate pH6/microwave (Dako, Santa Clara, California, USA). Sections were incubated with primary antibody anti-MDA (1:2,500; Abcam, Cambridge, UK) and anti-CD36 (1:200; BD, Franklin Lakes, New Jersey, USA) at 4°C overnight (see Appendix Table S7). After washing, sections were incubated with the fluorochrome-conjugated antibody goat anti-rabbit IgG Alexa 488 1:200 (Alexa Fluor, Waltham, Massachusetts, USA) for 1 h at room temperature (RT) in the dark. Nuclear staining was performed by incubating sections with Hoechst 33342 (1:1,000; Life Technologies, Carlsbad, California, USA). Sections were observed by Zeiss Axio Observer.Z1, with Apotome technology, and images were acquired with the software AxioVision Rel. 4.8. For each explanted heart subject, three consecutive slices and at least five fields for each slice were examined, excluding autofluorescence and aspecific signals. For ACM biopsy samples, all the samples were sliced and examined.

## C-MSC isolation and culture

C-MSCs were isolated and cultured as previously reported (Sommariva *et al*, 2016; Pilato *et al*, 2018). Briefly, ventricular samples were washed with PBS, cut into 2–3 mm pieces, and incubated at 37°C for 1.5 h under continuous agitation in Iscove's modified Dulbecco's media (IMDM; Gibco, Waltham, Massachusetts, USA) containing 3 mg/ml collagenase NB4 (Serva, Heidelberg, Germany). The digested solution was then centrifuged at 400 *g* for 10 min, washed with PBS, and centrifuged again. The obtained pellet was resuspended in GM, consisting of IMDM supplemented with 20% fetal bovine serum (FBS; Euroclone, Milan, Italy), 10 ng/ml basic fibroblast growth factor (R&D Systems, Minneapolis, Canada), 10,000 U/ml penicillin (Invitrogen, Carlsbad, California, USA), 10,000 µg/ml streptomycin (Invitrogen, Carlsbad, California, USA), and 20 mmol/l L-Glutamine (Sigma-Aldrich, St. Louis, Missouri, USA). The cells were seeded onto uncoated Petri dishes (Corning, Corning, New York, USA). Non-adherent cells were removed after 24 h.

The use of different C-MSC samples for different experiments is detailed in Appendix Table S6.

### C-MSC immunofluorescence analysis

C-MSC were plated on 1.8 cm$^2$ chamber slides (Thermo Fisher Scientific, Waltham, Massachusetts, USA) at a density of 20,000 cells/cm$^2$. After 24 h of culture in basal conditions, C-MSC were washed with PBS and fixed in 4% paraformaldehyde in PBS. After the blocking step in 10% goat serum (Sigma-Aldrich, St. Louis, Missouri, USA), cells were incubated with primary antibodies anti-MDA (1:2,500; Abcam, Cambridge, UK) and anti-CD36 (1:200; BD, Franklin Lakes, New Jersey, USA) at 4°C overnight. After washing, sections were incubated with the fluorochrome-conjugated antibody goat anti-rabbit IgG Alexa 488 1:200 (Alexa Fluor, Waltham, Massachusetts, USA) for 1 h at RT in the dark. Nuclear staining was performed by incubating sections with Hoechst 33342 (1:1,000; Life Technologies, Carlsbad, California, USA). Sections were observed by Zeiss Axio Observer.Z1, with Apotome technology, and images were acquired with the software AxioVision Rel. 4.8. For each dish, 15 fields were examined.

### C-MSC treatments

The medium used to prompt the adipogenic differentiation of C-MSC consists of IMDM supplemented with 10% FBS (Euroclone, Milan, Italy), 0.5 mmol/l 3-isobutyl-1-methylxanthine (Sigma-Aldrich, St. Louis, Missouri, USA), 1 µmol/l hydrocortisone (Sigma-Aldrich, St. Louis, Missouri, USA), 0.1 mmol/l indomethacin (Sigma-Aldrich, St. Louis, Missouri, USA), 10,000 U/ml penicillin (Invitrogen, Carlsbad, California, USA), 10,000 µg/ml streptomycin (Invitrogen, Carlsbad, California, USA), and 20 mmol/l L-Glutamine (Sigma-Aldrich, St. Louis, Missouri, USA). For the preparation of AM, refer to (Pilato *et al*, 2018).

C-MSC were plated in AM at a concentration of 20,000 cells/cm$^2$ and treated with 150 µg/ml oxLDL (see the paragraph "oxLDL preparation") (Asmis & Begley, 2003; Hamel *et al*, 2008; Lee *et al*, 2010), 20 µg/ml 13HODE (Cayman Chemicals, Ann Arbor, Michigan, USA) (Nagy *et al*, 1998; Kim *et al*, 2013), or 5 mmol/l NAC (Sigma-Aldrich, St. Louis, Missouri, USA) (Zolkipli *et al*, 2011; Wang *et al*, 2015; Mao *et al*, 2016; Pieralisi *et al*, 2016), a known thiolic antioxidant (Sunitha *et al*, 2013). After 72 h, treatment effects were evaluated by ORO (Sigma-Aldrich, St. Louis, Missouri, USA) staining and Western blotting analysis.

### C-MSC lipid staining

C-MSC were plated at a concentration of 20,000 cells/cm$^2$ and cultured in AM. Fat accumulation was tested by ORO (Sigma-Aldrich, St. Louis, Missouri, USA) intracellular lipid staining. In detail, C-MSC were stained with 1% ORO (Sigma-Aldrich, St. Louis, Missouri, USA) solution in 60% isopropanol for 1 h after 5-min fixation with 4% paraformaldehyde in PBS. After five washes in PBS to ensure the removal of unbound dye, quantitative results were obtained by evaluating luminance in the 255 red channel with the ImageJ program (at least 15 fields were evaluated per condition per patient).

### OxLDL preparation

Fresh plasma purchased from healthy donors (Niguarda Ca' Granda Hospital, Milano) was brought to a density of 1.019 g/ml with KBr and subsequently centrifuged at 40,000 rpm for 16 h to remove VLDL particles. After this operation, the gradient was adjusted to a density of 1.063 g/ml and samples centrifuged for further 24 h at the same speed. LDL were then isolated, dialyzed for 48 h at 4°C, sterilized by filtration (0.22 µm; Millipore, Burlington, Massachusetts, USA) and then characterized for their protein and cholesterol content. Aliquots of 2–3 ml were oxidized by addition of an equal volume of $CuSO_4$ (final concentration 2.7 mM) for 18 h, under stirring. Oxidation was documented by agarose gel electrophoresis and by gas-liquid chromatography analysis, by which we monitored the specific disappearance of polyunsaturated fatty acids, namely linoleic, arachidonic, and eicosapentaenoic ones. OxLDL were then utilized for cell culture experiments, under sterile conditions.

### OxLDL internalization assay in C-MSC

C-MSC were plated on 1.8 cm$^2$ chamber slides (Thermo Fisher Scientific, Waltham, Massachusetts, USA) at a density of 20,000 cells/cm$^2$ either in GM or in AM for 3 days. 10 µg/ml DiI-oxLDL (Thermo Fisher Scientific, Waltham, Massachusetts, USA) were added. After 3 h, the cells were fixed for 5 min in 4% paraformaldehyde in PBS, stained with Hoechst 33342 (1:1,000; Life Technologies, Carlsbad, California, USA), and the slides mounted. Pictures were acquired with Zeiss Axio Observer.Z1, with Apotome technology. For the quantification of intracellular DiI, other cells from the same cultures were treated as described above and acquired with FACS Gallios (Beckman Coulter, Brea, California, USA).

### *PKP2* silencing

HC C-MSC were plated at a density of 12,500 cell/cm$^2$ in growth medium and transduced with pooled lentiviral particles containing shRNAs targeting both variants of human *PKP2* (Gene ID 5381) in psi-LVRU6GP (with U6 promoter, eGFP reporter, puromycin resistance; Genecopoeia; Rockville, Maryland) or with the correspondent scrambled control lentiviral particles (Genecopoeia; Rockville, Maryland) for 24 h. After checking the transduction efficiency by detection of the GFP signal, 2 µg/ml puromycin was added to select transduced cells.

After cell amplification, PKP2 reduction was assayed by Western blot. Scrambled control and *PKP2* shRNA C-MSC were plated in AM at a concentration of 20,000 cells/cm$^2$ and treated with 150 µg/ml oxLDL. After 72 h, treatment effects were evaluated by Oil Red O (ORO; Sigma-Aldrich St. Louis, Missouri, USA) staining.

### CD36 knockdown in C-MSC

C-MSC were cultured for 24 h in low-serum medium without antibiotics (IMDM, 2% FBS (Euroclone, Milan, Italy) and 20 mmol/l L-Glutamine (Sigma-Aldrich, St. Louis, Missouri, USA). 0.05 µM of Human Silencer Select Pre-designed CD36 siRNA (4392422-S2646 siRNA; Life Technologies, Carlsbad, California, USA) or 0.05 µM Silencer Select Negative Control (4390844 scramble; Life Technologies, Carlsbad, California, USA) and 4 µl lipofectamine RNAiMAX

(Life Technologies, Carlsbad, California, USA) were added to 300 µl of Opti-MEM medium (Life Technologies, Carlsbad, California, USA). After 15 min at RT, the transfection reactions were added to the cells in 1.5 ml low-serum/no antibiotics medium. After 24 h, the medium was changed to AM supplemented with 150 µg/ml oxLDL for the following 72 h. CD36 reduction was confirmed by Western blot at the end of the experiment.

### PPARγ antagonism in C-MSC

C-MSC were cultured for 72 h in AM with 5 µM GW9662 (Sigma-Aldrich, St. Louis, Missouri, USA). The treatment was added to the medium every 8 h. To check oxLDL internalization, 10 µg/ml DiI-oxLDL (Thermo Fisher Scientific, Waltham, Massachusetts, USA) were added 3 h before the end of the experiment, following the protocol described above (see "OxLDL internalization assay"). Pictures were acquired with Zeiss Axio Observer.Z1, with Apotome technology, and images were acquired with the software AxioVision Rel. 4.8. For each biological sample, 15 fields were examined.

### Flow cytometry in C-MSC

To evaluate C-MSC oxidative status, cells cultured in basal medium were incubated for 30 min with 10 µM dichlorofluorescein (Sigma-Aldrich, St. Louis, Missouri, USA) and detached with TrypLE Select (Life Technology, Carlsbad, California, USA), and the conversion into the fluorescent dye 2′,7′-DCF by cell ROS was measured by flow cytometry (Gallios, Beckman Coulter, Brea, California, USA). The mean FITC fluorescence was measured.

To determine the correlation between CD36 expression and lipid accumulation, cells were stained using 12.5 ng/ml Nile Red (Invitrogen, Carlsbad, California, USA), to mark intracellular neutral lipids, and 2.5 µl of anti-CD36 antibody (Life Technologies, Carlsbad, California, USA). The mean of the fluorescence was determined for Nile Red and CD36 for each sample.

To quantify the DiI-oxLDL internalization, cells were treated with 10 µg/ml DiI-oxLDL for 3 h, detached with TrypLE Select (Life Technologies, Carlsbad, California, USA), and acquired with FACS Gallios (Beckman Coulter, Brea, California, USA). The mean APC fluorescence was determined for each sample.

### Western blot in C-MSC

Total proteins from C-MSC were obtained by Laemmli lysis buffer. After quantification with DC protein assay (Bio-Rad, Hercules, California, USA), proteins were run on SDS–PAGE gel (NUpage precast 4–12%; Invitrogen, Carlsbad, California, USA) and transferred to nitrocellulose membrane (Bio-Rad, Hercules, California, USA). The membrane was blocked in 5% skimmed milk-TBS for 1 h at RT and incubated overnight at 4°C with primary antibodies against GAPDH, PPARγ, and CD36 (see Appendix Table S7). After washes, the membrane was incubated for 1 h at RT with the appropriate HRP-conjugated secondary antibody goat anti-rabbit or goat anti-mouse (GE Healthcare, Chicago, Illinois, USA). Blots were washed and developed with the ECL system (Bio-Rad, Hercules, California, USA). Images were acquired with the Alliance Mini 2 M System (UVITEC, Cambridge, UK), and densitometric analysis was performed using Alliance Mini4 16.07 software (UVITEC,

Cambridge, UK). Data are normalized expressing as 1 the comparison group in order to highlight the fold differences between different groups or treatment.

### Glutathione quantification in C-MSC

Levels of GSH and GSSG were determined by a previously described and validated LC-MS/MS method (Squellerio *et al*, 2012). Briefly, cells cultured in GM have been washed twice with PBS, detached with trypsin, and then collected and centrifuged at 400 *g* for 10 min. The supernatant was removed, the pellet was resuspended in 50 µl of PBS, and proteins were precipitated with 50 µl of 10% trichloroacetic acid with the addition of 1 mmol/l EDTA and stored at −80°C until the analysis. Thawed samples were further diluted 1:10 with formic acid 0.1% before the LC-MS/MS analysis. The LC-MS/MS analysis was performed using an Accela HPLC System (Thermo Fisher Scientific, Waltham, Massachusetts, USA) coupled to a triple quadrupole mass spectrometer TSQ Quantum Access (Thermo Fisher Scientific, Waltham, Massachusetts, USA) outfitted with electrospray ionization source operating in positive mode. The chromatographic separation was conducted on a Luna PFP column (2.0 mm × 100 mm, particle size 3.0 µm, Phenomenex, Torrance, California, USA) maintained at 35°C. Analytes were eluted under isocratic conditions at 200 µl/min by 1% methanol in 0.75 mM ammonium formate adjusted to pH3.5 with formic acid. The analytes were detected by multiple reaction monitoring, and the transitions monitored (precursor ion > product-fragment ions) were $m/z$ 308.1 → $m/z$ 76.2, 84.2, 161.9 (GSH) and $m/z$ 613.2 → $m/z$ 230.5, 234.6, 354.8 (GSSG). A linear 6-point calibration curve (range 0.25–8 µM for GSH and 0.008–0.25 µM for GSSG) was used for the quantification.

### Analysis of C-MSC lipids

Cell lipids were extracted by hexane/isopropanol 3:2, plus butylated hydroxytoluene 0.005% as antioxidant. Known amounts of proper internal standards (stigmasterol, cholesteryl heptadecanoate, triheptanoin, nonadecanoic acid; Sigma-Aldrich, St. Louis, Missouri, USA) were added for the analysis of free cholesterol (FC) and esterified cholesterol (CE), triglycerides (TG), and free fatty acids (FFA), respectively. Lipid extracts were dried in a steam of nitrogen, aliquoted, and conserved at −80°C in the dark, until use.

An aliquot was loaded onto pre-run and activated channeled Silica TLC plates (BioMap) and run in hexane-diethyl ether-acetic acid (80:20:1 vol/vol/vol). The plates were then sprayed with dichlorofluorescein (0.15% in ethanol), and the spots corresponding to those of FC, FFA, TG, and CE standards were identified by UV light and scraped out of the TLC.

Samples were analyzed by a gas-liquid chromatographer (GLC 1000; DANI Instruments, Cologno Monzese, Italy) equipped with an autosampler HT300A (HTA, Brescia, Italy), a fused silica column (MEGA-5 30 m length, 0.3 mm diameter, 0.15 µm film thickness; Mega Columns, Legnano, Italy) and a flame ionization detector. Hydrogen flow was at a constant pressure of 1.2 bar. The oven temperature was constant (260°C, 8 min run) for FC, while ranged from 120°C to 300°C for FFA, TG, and CE (total run 45 min).

FC was resuspended in hexane/isopropanol and analyzed without derivatization. The other lipid classes were processed with

methanolic acid 3N at 30–80°C for 30–120 min and analyzed for their fatty acid content. Peaks were identified by comparing their retention times with those of standard and their area determined by a dedicated software (Clarity). To calculate the total mass of each lipid class, the areas of all the peaks corresponding to the fatty acids were summed and the real mass determined by comparison with the area of the internal standards. Results were normalized by the number of cells in each dish ($\mu$g lipid/$10^6$ cells).

## Generation of ACM and HC hiPSC

hiPSC from one ACM patient, carrying the deletion of the whole *PKP2* exon 4 leading to a predicted truncated protein (p.N346Lfs*12), and one HC of the same family were obtained and characterized as previously described (Ermon *et al*, 2018; Meraviglia *et al*, 2018). Briefly, every hiPSC line has been generated using episomal vectors carrying OCT3/4, SOX2, KLF4, and L-MYC and has been subsequently grown on a mouse embryonic fibroblast (MEF) feeder layer. Clones that passed the quality check were adapted at passage 10–15 to grow in feeder-free condition (without MEF) directly onto six-well plates coated with Matrigel® matrix (Corning, Corning, New York, USA) and cultured in the commercially available xeno-free medium StemMACSTM iPS-Brew XF (Miltenyi Biotec, Bergisch Gladbach, Germany).

## Cardiomyogenic differentiation of hiPSC

hiPSC were propagated on Matrigel® (Corning, Corning, New York, USA)-coated plates, and the cardiomyogenic differentiation was performed using PSC Cardiomyocyte Differentiation Kit (Thermo Fisher Scientific, Waltham, Massachusetts, USA). After 30 days, cardiomyocytes were dissociated at single cells using Multi Tissue Dissociation Kit 3 (Miltenyi Biotec, Bergisch Gladbach, Germany) and enriched by magnetic separation with the QuadroMACS™ Separator and PSC-Derived Cardiomyocyte Isolation Kit (Miltenyi Biotec, Bergisch Gladbach, Germany). Cells were then maintained in culture for additional 25 days in the following different conditions: growth medium (composed by High Glucose DMEM (Gibco, Waltham, Massachusetts, USA), 2% of Hyclone Fetal Bovine Defined (GE Healthcare, Chicago, Illinois, USA), 1% of Non-Essential Amino Acids, Penicillin/Streptomycin and 0,09% of β-mercaptoethanol), and Adipogenic Medium (growth medium supplemented with 50 μg/ml Insulin (Sigma-Aldrich, St. Louis, Missouri, USA), 0.5 μM Dexamethasone (Sigma-Aldrich, St. Louis, Missouri, USA), 0.25 mM 3-isobutyl-1 methylxanthine (IBMX) (Sigma-Aldrich, St. Louis, Missouri, USA), 200 μM Indomethacin (Sigma-Aldrich, St. Louis, Missouri, USA) with 5 μM Rosiglitazone (Vinci-Biochem, Vinci, Italy). The culture medium was changed every other day. Cells were analyzed through immunofluorescence and FACS analysis.

## hiPSC-CM immunofluorescence analysis

Cells were fixed with 4% paraformaldehyde for 15 min and then permeabilized (PBS with 0,1% Triton X100) for 10 min at RT. Cells were then blocked in PBS with 5% Goat serum for 1 h at RT and incubated with anti-αSARC overnight at 4°C. After washing, cells were incubated with the proper secondary antibody for 1 h at 37°C.

Intracellular lipid droplet accumulation was evaluated using BODIPY 493/503 assay (dilution 0.1 μg/ml in PBS; Thermo Fisher Scientific, Waltham, Massachusetts, USA) incubated for 20 min at RT and nuclei were stained with DAPI (Invitrogen, Carlsbad, California, USA). The images were acquired using confocal microscopy (Leica Microsystem CMs GmBH Type: TCSSP8X) and analyzed with ImageJ software. Intensity fluorescence was normalized on nuclei number.

## hiPSC-CM flow cytometry analysis

Cells were dissociated at single cell using Multi Tissue Dissociation Kit 3 (Miltenyi Biotec, Bergisch Gladbach, Germany), following the manufacturer's instructions and blocked in FACS buffer (PBS containing 0.5% FBS and 2 mM EDTA). Cells were incubated with anti-CD36 for 15 min at 4°C avoiding direct light and then washed and resuspended with FACS buffer. Cardiomyocytes were identified and gated based on their forward and side scatter using the S3Cell Sorter (Bio-Rad, Hercules, California, USA); median fluorescence intensity (MFI) of PE was calculated on the gated cells. Data were analyzed using FlowJo software. Median intensity fluorescence is presented as MFI(PE) sample – MFI(PE) isotype.

## ACM murine model: *Pkp2*+/− mice

C57Bl/6 *Pkp2* heterozygous knock-out mice (*Pkp2*+/−) were produced by Prof. Birchmeier, as described (Grossmann *et al*, 2004). The homozygous mice were embryonic-lethal, while heterozygous mice were healthy and fertile. For our experiments, we used 28 C57Bl/6 *Pkp2*+/− mice and 28 siblings C57Bl/6 wild type (WT), as control. This number is the result of a power analysis by means of G*Power 3.1.9.2 software. We used a two-way ANOVA test, taking into account the interaction between the differences due to the different strains to be used, and that due to the diets. Thanks to previous results with the same murine model, given the expected differences in cardiac lipid accumulation and the expected inter-group standard deviation, we obtained an effect size f of 0.52. The calculation of sample size showed that ˜9 animals/group/treatment are required for an 85% probability of demonstrating differences from the mean, with a *P* value of 0.05.

Experiments were authorized on 27/07/2015 by the Italian Ministry of Health, protocol no. 779/2015-PR.

## Murine C-MSC isolation and culture

Explanted hearts from 5 WT and 5 *Pkp2*+/− mice (age: 10 weeks) were washed with PBS, cut into 2–3 mm pieces, and incubated at 37°C for 1 h under continuous agitation in IMDM containing 3 mg/ml collagenase NB4. After a PBS wash, the pellet was resuspended in GM and seeded. Non-adherent cells were removed after 24 h. AM medium was used to prompt the adipogenic differentiation of murine C-MSC for 6 days.

## High-fat diet

10 WT and 10 *Pkp2*+/− mice (age: 10 weeks) were fed for 3 months with a high-cholesterol (279.6 mg/kg) and high-fat (60% kcal) diet (HFD; OpenSource DIETS, New Brunswick, New Jersey, USA) (Khan-Merchant *et al*, 2002).

9 WT and 9 *Pkp2*+/− mice were fed with chow diet (CD; Open-Source DIETS, New Brunswick, New Jersey, USA).

Heart volumes and functionality were assessed before the beginning of the diet and every month by echocardiography. Bodyweight was monitored every month. Blood samples were taken before and after the diet. At sacrifice, hearts were explanted after perfusion with saline solution. Sections of heart specimens were used for lipid accumulation analysis, PPARγ, MDA, and CD36 immunofluorescence staining. Protein extracts from total tissue lysates were used to evaluate PPARγ, MDA, and CD36 expression through Western blot analysis.

## High-fat diet plus atorvastatin

Nine WT and 9 *Pkp2*+/− mice (age: 10 weeks) were fed for 3 months with a HFD (D12492, Open Source, New Brunswick, New Jersey, USA) to which 20 mg/kg atorvastatin (Khan *et al*, 2018; Peng *et al*, 2018) was added. Heart volumes and functionality were assessed before the beginning of the diet and every month by echocardiography. Bodyweight was monitored every month. Blood samples were taken before and after the diet. At sacrifice, hearts were explanted after perfusion with saline solution. Sections of heart specimens were used for lipid accumulation analysis, PPARγ, MDA, and CD36 immunofluorescence staining. Protein extracts from total heart tissue lysates were used to evaluate PPARγ, MDA, and CD36 expression through Western blot analysis.

## Echocardiographic and electrocardiographic analyses

Transthoracic echocardiography was performed using the Vevo2100 high-resolution imaging system (VisualSonics, Toronto, Canada) and a 40-MHz linear transducer with simultaneous electrocardiographic recording, as previously reported (Milano *et al*, 2014). Analyses were performed on mice lightly anesthetized with 0.5% to 1% isoflurane (heart rate: 480–550 beats/min) at the following timepoints: 1 day before starting the diet (pre-diet) and on 1, 2, and 3 months of CD, HFD or HFD+atorva.

RV parameters in systole and diastole were acquired from a parasternal long-axis view and measured from images acquired in M mode, using the depth interval (in mm) generic measurements tool (Urboniene *et al*, 2010; Seta *et al*, 2011; Hansmann *et al*, 2012).

Two-dimensional short-axis M-mode echocardiography was performed at the level of the midpapillary muscle to measure LV parameters, in systole and diastole.

All measurements were averaged from a minimum of three cycles during diastole and systole corresponding to the electrocardiogram. Data and imaging were analyzed using the VisualSonics Cardiac Measurements Package by a blinded investigator. Parameters were normalized on mice heart weight.

Surface electrocardiographic signal (lead II via limb electrodes) were acquired in all mice during echocardiography (lightly anesthetized mice). CSV files of each mouse physiological data were exported, and analyzed in Excel. Specifically, QRSp (as defined in (Merentie *et al*, 2015)), terminal activation duration (TAD) and QRS amplitude were measured in at least 10 consecutive beats for each animal by a blinded investigator.

## Murine blood sampling

Mice were anesthetized with 4% isoflurane and maintained asleep with 1% isoflurane. The blood sampling was performed through tail vein using 25G needle, after tail pre-heating and local application of anesthetic.

Whole blood was collected into EDTA-coated tubes (Fisher Scientific, Waltham, Massachusetts, USA). Separated plasma was obtained after centrifugation for 15 min at 2,000 g at 4°C and stored at −80°C until the analysis.

## Total cholesterol distribution in lipoproteins from mouse plasma

50 µl of mouse plasma has been injected twice in a HPLC system (Jasco 920; Jasco, Cremella, Italy), with a pre-column (6 mm ID × 4 cm; Tosoh, Tokyo, Japan), two consecutive inverse phase, size-exclusion, anionic-exchange TSK-GEL LIPOPROPAK XL (7.8 mm ID × 30 cm) columns (Tosoh, Tokyo, Japan) and with a UV-VIS detector (Jasco, Cremella, Italy). Readings were performed at a 280 nm. The mobile phase consisted of 6.9 g of sodium monobasic phosphate, 50 mg BRIJ, and 30 ml isopropanol per liter (pH8).

We divided each sample into 50 fractions (one every minute), and we collected those from #22 to #45, corresponding to VLDL, IDL, LDL, and HDL. Each obtained fraction (1 ml) was frozen at −80°C, lyophilized, and then reconstituted with 200 µl of water. On portions of these samples, total cholesterol was measured by commercial colorimetric kits (ABX Pentra, Roma, Italy) at 490 nm by a spectrophotometer (Bio-Rad, Hercules, California, USA).

## oxLDL in murine plasma samples

For the quantitative determination of oxLDL in murine plasma samples before and after the HFD or HFD+atorva, Mouse Oxidized Low Density Lipoprotein ELISA kit was used (CUSABIO, Houston, Texas, USA), following the manufacturer's instructions. Absorbance was determined with a spectrophotometer (Berthold Technologies, Bad Wildbad, Germany) at 450 nm. Results were inferred according to a calibration curve constructed using the standards provided in the kit.

## Histological characterization of *Pkp2*+/− hearts

Explanted hearts from WT and *Pkp2*+/− mice, in CD, HFD, and HFD+atorva conditions, were fixed in 4% paraformaldehyde (Santa-Cruz, Dallas, Texas, USA) overnight at 4°C. The following day, after three washes in PBS, the hearts were transferred in 15% sucrose in distilled water overday at RT and then in 30% sucrose in distilled water overnight at 4°C. The explanted hearts were embedded in OCT (Thermo Fisher Scientific, Waltham, Massachusetts, USA) and stored at −80°C until use. After sectioning with a cryostat (Thermo Fisher Scientific, Waltham, Massachusetts, USA), OCT embedded sections (6 µm thick) were de-frosted and washed twice in PBS. The immunofluorescence analysis was performed following blocking in 2% goat serum (Sigma-Aldrich, St. Louis, Missouri, USA) for 30 min. Sections were incubated with primary antibodies (see Appendix Table S7) anti-PPARγ (1:200; Life Technologies, Carlsbad, California, USA), anti-MDA (1:2,500; Abcam,

Cambridge, UK), anti-CD36 (1:200, Neuromics, Edina, Minneapolis, USA), anti-CD105 (1:40, R&D System, Minneapolis, Minnesota, USA), anti-troponin T (1:100, Abcam, Cambridge, UK), anti-perilipin1 (1:100, OriGene, Rockville, Maryland, USA), and anti-CX43 (1:400, Abcam, Cambridge, UK) at 4°C overnight. After washing, sections were incubated with the fluorochrome-conjugated antibodies goat anti-rabbit IgG Alexa 488 (1:200; Alexa Fluor, Waltham, Massachusetts, USA), Streptavidin-Alexa Fluor 594 (1:200; Alexa Fluor, Waltham, Massachusetts, USA), goat anti-rabbit 546 (1:200; Alexa Fluor, Waltham, Massachusetts, USA), and goat anti-guinea pig 488 (1:200; Alexa Fluor, Waltham, Massachusetts, USA) for 1 h at RT in the dark. Nuclear staining was performed by incubating sections with Hoechst 33342 (1:1,000; Life Technologies, Carlsbad, California, USA). Sections were observed with Zeiss Axio Observer.Z1, with Apotome technology, and images were acquired with the software AxioVision Rel. 4.8. For each explanted heart, at least 10 fields for five consecutive transversal slices were quantified.

For the lipid accumulation analysis, the sections were stained with ORO (Sigma-Aldrich, St. Louis, Missouri, USA) for 1 h at RT and then washed five times in PBS to ensure the removal of the aspecific dye. Quantitative results were obtained by evaluating red area vs. total tissue area (all the section surface of five consecutive slices for each sample was quantified).

### Western blot in murine total heart tissue

Total proteins from murine total heart tissue lysates were obtained by Laemmli lysis buffer. After quantification with DC protein assay (Bio-Rad, Hercules, California, USA), proteins were run on SDS–PAGE gel (NUpage precast 4–12%; Invitrogen, Carlsbad, California, USA) and transferred to nitrocellulose membrane (Bio-Rad, Hercules, California, USA). The membrane was blocked in 5% skimmed milk-TBS for 1 h at RT and incubated overnight at 4°C with primary antibodies against GAPDH, PPARγ, MDA, and CD36 (see Appendix Table S7). After washes, the membrane was incubated for 1 h at RT with HRP-conjugated secondary antibody goat anti-rabbit (GE Healthcare, Chicago, Illinois, USA). Blots were washed and developed with the ECL system (Bio-Rad, Hercules, California, USA). Images were acquired with the Alliance Mini 2 M System (UVITEC, Cambridge, UK), and densitometric analysis was performed using Alliance Mini4 16.07 software (UVITEC, Cambridge, UK). Data are normalized expressing as 1 the comparison group in order to highlight the fold differences between different groups or treatment.

### Patient fat infiltration analysis at MRI

Intramyocardial fat mass was evaluated as previously described (Aquaro et al, 2012; Aquaro et al, 2014). Briefly, steady-state free procession (SSFP) were used and the following acquisition parameters were applied: 30 phases, 10–25 views per segment, NEX 1, FOV 40 cm, a matrix of 224 x 224, a 60° flip angle, TR 3.6–4.2 and TE = TR/2. Images were acquired using a 1.5-T unit (Discovery MR450, GE Healthcare, Milwaukee, Minneapolis, USA). Fat in the ventricular myocardium appears as a hyperintense area surrounded by a hypointense band, the so-called chemical shift artifact (Aquaro et al, 2014). Manual contouring of the chemical

### The Paper Explained

**Problem**

Arrhythmogenic cardiomyopathy (ACM) is a genetically determined heart condition. It is hallmarked by a gradual fibro-adipose replacement of the ventricular myocardium, heart failure, malignant arrhythmias, and sudden death. Few information is available regarding the mechanisms mediating the phenotypic variability among carriers of the same ACM mutation. In addition, no pharmacological approaches are available in the clinical practice to counteract cardiac adipogenic substitution.

**Results**

We demonstrated that oxLDL increase ACM adipogenesis through a mechanism implying over-activation of PPARγ, the main effector of lipid accumulation. This novel pathogenic mechanism was investigated in patients and verified with in vitro experiments on two cardiac cell types and in vivo studies. Importantly, antioxidants and atorvastatin treatments counteracted ACM phenotypes.

**Impact**

This study adds a piece to the puzzle of phenotypic variability among carriers of the same ACM mutation. Increased oxLDL plasma levels may represent both a precision medicine tool to identify patients at high disease burden and a target of therapeutic approaches. A proof of principle is given of the efficacy of a new treatment with atorvastatin and antioxidants in counteracting oxidized lipid-dependent cardiac adipogenesis and related dysfunction. Future drug repositioning studies will compare the efficacy of such treatments in mitigating ACM phenotypes.

shift artifact (including the black contour) was drawn, and its extent was measured.

### Statistical analysis

Discrete variables were analyzed with Fisher's exact test. Continuous variables were reported as mean±standard error. Comparisons between normally distributed groups were performed using either paired or unpaired two-tailed Student's t-tests, whereas populations without a Gaussian distribution were compared using Mann–Whitney tests. The $n$ indicated in each figure legend corresponds to biological replicates. Comparisons among three or more groups were performed with one-way or two-way ANOVA test, in association with Bonferroni multiple comparison post-tests. When both intragroup and intergroup values are tested, red lines and asterisks identify test differences within the same group, while intergroup differences are in black. X-Y correlation analyses have been determined. Comparison of slopes of linear regressions was performed with the following method: $t = (b1–b2) / sb1,b2$ where b1 and b2 are the two slope coefficients and sb1,b2 the pooled standard error of the slope. Receiver operating characteristic (ROC) plot was used to determine the oxLDL level cut-off value minimizing the difference between sensitivity and specificity in discriminating ACM patients vs. HC individuals. Kaplan–Meier curves were performed to determine the actual risk of MAE and analyzed through log-rank (Mantel–Cox) test. Statistics were performed using GraphPad Prism software. Results were considered statistically significant for $P$ values < 0.05 (see Appendix Table S8 for the details). Power analyses, for both in vitro and in vivo experiments, were performed using G*Power 3.1.9.2 software.

## Data availability

This study includes no datasets deposited in external repositories.

**Expanded View** for this article is available online.

## Acknowledgements

This work was supported by the Telethon Grant GGP16001 to Prof. Pompilio, Dr. Rossini, and Prof. Corsini, by the Italian Ministry of Health Ricerca Finalizzata Grant GR-2016-02362024 to Dr. Elena Sommariva, and by the Department of Innovation, Research and Universities of the Autonomous Province of Bolzano-South Tyrol (Italy). We express our gratitude to the patients and their families for their participation in this study. Thanks to Prof. Delmar for providing the *Pkp2+/−* mouse model, to Dr. Agnese Granata, Dr. Anna Guarino, Dr. Silvia Brambilla, Dr. Fabio Cattaneo, and Dr. Benedetta Ermon for technical help, and to Prof. Viviana Cavalca for critical reading of the manuscript.

## Author contributions

ES, GP, and IS conceived and designed the research, with the help of AC, AR, and CT. MiCa, VC, ADR, CC, CT, and WR collected, analyzed, and interpreted clinical data. EC and AP provided samples from control donors. ES, IS, LA, SDM, VM, MDM, CV, GC, LT, BP, and MaCa performed *in vitro* studies. DA, SM, and EC performed magnetic resonance and collected and analyzed the data. WB generated the *Pkp2+/−* murine model. GM, AS, MC, and IS performed *in vivo* experiments. EK, AR, and MP analyzed genetic data. MaCh performed statistical analysis. IS, ES, and GP wrote the manuscript. ES and GP supervised the project. All authors critically revised the manuscript.

## Conflict of interest

The authors declared that they have no conflict of interest.

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
