## [Review Process File · EMBO Molecular Medicine]

Oxidized LDL-dependent pathway as new pathogenic trigger in Arrhythmogenic Cardiomyopathy.

Elena Sommariva, Ilaria Stadiotti, Michela Casella, Valentina Catto, Antonio Dello Russo, Corrado Carbucicchio, Lorenzo Arnaboldi, Simona De Metrio, Giuseppina Milano, Alessandro Scopece, Manuel Casaburo, Daniele Andreini, Saima Mushtaq, Edoardo Conte, Mattia Chiesa, Walter Birchmeier, Elisa Cogliati, Adolfo Paolin, Eva König, Viviana Meraviglia, Monica De Musso, Chiara Volani, Giada Cattelan, Werner Rauhe, Linda Turnu, Benedetta Porro, Matteo Pedrazzini, Marina Camera, Alberto Corsini, Claudio Tondo, Alessandra Rossini, and Giulio Pompilio

DOI: [10.15252/emmm.202114365](https://doi.org/10.15252/emmm.202114365)

Corresponding authors: Elena Sommariva (esommariva@cclf.it)

Review Timeline:

Submission Date:	15th Apr 20
Editorial Decision:	20th May 20
Revision Received:	6th Apr 21
Editorial Decision:	7th May 21
Revision Received:	7th Jun 21
Editorial Decision:	28th Jun 21
Revision Received:	6th Jul 21
Accepted:	6th Jul 21

Editor: Lise Roth

Transaction Report:

Dear Dr. Sommariva,

Thank you for the submission of your manuscript to EMBO Molecular Medicine. We have now received feedback from two of the three reviewers who agreed to evaluate your manuscript. Given that both referees provide very similar recommendations, we prefer to make a decision now in order to avoid further delay in the process.

As you will see from the enclosed reports, neither referee supports publication of the manuscript in EMBO Molecular Medicine. They both acknowledge that the study is of potential interest, however, they are not convinced that as it stands the main conclusions of the study are well supported by the data. Considering the substantial points raised and the overall rather low level of support provided by the reviewers, I am afraid I see little choice but to return the manuscript to you at this point with the decision that we cannot offer to publish it.

Given the potential interest of the findings, we would, however, be willing to consider a new manuscript on the same topic if at some time in the near future you obtained data that would considerably strengthen the message of the study and address the referees' concerns in full. To be completely clear, however, I would like to stress that if you were to send a new manuscript, this would be treated as a new submission rather than a revision and would be reviewed afresh, in particular with respect to the literature and the novelty of your findings at the time of resubmission. If you decide to follow this route, please make sure you nevertheless upload a letter of response to the referees' comments.

I am sorry that I could not bring better news this time and hope that the referee comments are helpful in your continued work in this area.

Yours sincerely,

Lise Roth

Lise Roth, Ph.D
Editor
EMBO Molecular Medicine

***** Reviewer's comments *****

Referee #2 (Comments on Novelty/Model System for Author):

The findings presented here extend the well described involvement of PPAR signaling for other metabolic related disease to ACM. Further, the use of statins to attenuate incidences of ventricular arrhythmias has been previously described. The model systems used for experimentation are appropriate.

Referee #2 (Remarks for Author):

The manuscript by Sommariva et al. describes an oxLDL/CD36/PPAR circuitry axis in which elevations in oxLDL cellular uptake, driven by CD36 up-regulate PPAR expression and activity. In a feed forward circuit, the authors describe such phenomenon as a potential mechanism contributing to lipid accumulation in ACM affected cells. Moreover, the authors translate their in vitro findings using a genetic mouse model and demonstrate the utility of cholesterol lowering pharmacology to alleviate the disease phenotype. The findings reported here are a great advancement to understanding the (patho)biology of ACM. However, there are several limitations and concerns that warrant further consideration.

Major concerns:

- 1) The authors should explain why they observe significantly more ROS and lipid peroxidation in C-MSCs in ACM vs HC cells, but no difference in cellular oxidant defense mechanisms. Moreover, the authors should also describe why the addition of the antioxidant NAC decreased lipid accumulation (as concluded to be driven by oxidant stress) in ACM C-MSCs, yet those cells (based on the authors' data) do not appear to have any defects in cellular antioxidant defense.
- 2) The authors should make a better attempt to elucidate a mechanism by which relatives carrying the ACM mutation do not display the ACM phenotype. It is also not clear whether oxLDL is the consequence of or the cause of the ACM phenotype.
- 3) Given the susceptibility of this disease model to cardiac arrhythmias, the authors need to characterize the electrical activity in their mouse model with programmed electrical stimulation.
- 4) The mechanistic work dissecting the oxLDL/CD36/PPAR circuitry could be better described. For example, the difference in CD36 expression is marginal and it would be expected that silencing CD36 expression would preclude entry of oxLDL into the cell. Moreover, the mechanistic work describing the biological function of PPAR should be presented as a main figure, as this is central to the work reported here.
- 5) Figure 2D - The authors should represent an uncut gel, or where splice and with more representative lanes. It's unclear how the ratio of POI/GAPDH is ~ 1.0 when clearly the density of the POIs are much less than GAPDH. Importantly, it does not look apparent that any difference in the POIs exist between groups. Given the authors attempt to correlate increased CD36/PPAR expression in ACM affected cells, this should be significant attention. Moreover, if the authors used numerical transformation for control/experimental normalization, this should be described in the methods section.
- 6) Given that HC hiPSC-CMs are immune to lipid accumulation under AC + PPAR agonistic conditions, whereas ACM hiPSC-CMs are not might suggest dysfunctional lipid metabolism. The authors should explore this using mitochondrial functional assays (Seahorse).
- 7) It's important to not only describe the accumulation of lipids in ACM affected cells, but also how it relates to the pathological consequence to the disease.
- 8) The authors may consider using their mouse model to identify cell populations in the heart responsible for fatty tissue deposition.

Minor concerns:

- 1) The manuscript should be professionally proofread for English grammar.
- 2) Individual data points for bar graphs should be presented.
- 3) Figure 1C - with the exception of a couple potential outliers, it doesn't appear that there is any correlation between scar and oxLDL. It's interesting that $p=0.03$. The authors should report R squared with regression statistics.
- 4) Figures 1D,E - Normalizing to nuclei is not appropriate for this measurement. There are apparent differences in nuclei counts between the two images, which largely skews the data (especially in figure 1D. Further, there appears to be autofluorescence. Please describe how this was corrected for data normalization. A better method for MDA histology be colorimetric microscopy using agents such as DAB.
- 5) Figure 3C - Lanes on the representative gel should be labeled.
- 6) Reporting of non-significant data as a trend should be minimized.
- 7) My comment #4 should be extended to all figures using that methodology.

Referee #3 (Comments on Novelty/Model System for Author):

This paper addressed an interesting topic. The authors hypothesize that an oxLDL/CD36/PPAR γ circuitry triggers adipogenesis in arrhythmogenic cardiomyopathy. However, I believe key data is missing to support this hypothesis, particularly evidence that oxLDL is actually been taken up into the cardiomyocytes of ACM patients. Other experimental data needed to support the conclusions is also missing. These and other concerns are discussed in detail in the "Comments to Authors".

Referee #3 (Remarks for Author):

This study addressed the hypothesis that oxLDL uptake into cardiomyocytes, via CD36, activates PPAR γ and triggers adipogenesis in arrhythmogenic cardiomyopathy (ACM). ACM patients showed higher levels of oxLDL and 13HODE compared to healthy controls. In ACM patient cardiac-mesenchymal stromal cells (C-MSCs) oxLDL internalization through CD36 resulted in PPAR γ up-regulation and lipid accumulation. In Pkp2 +/- mice a high fat diet increased cardiac adipogenesis, which was prevented by atorvastatin-treatment. It is concluded that altered oxidized lipid metabolism and oxidative stress increase oxLDL bioavailability, which is internalized by C-MSC by CD36 receptor, thus acting through 13HODE-mediated PPAR γ activation to promote cardiac adipogenic differentiation.

General Comments:

This paper addresses the intriguing hypothesis that cardiomyocyte oxLDL internalization via CD36 can promote adipogenesis in ACM. The authors do show that plasma levels of oxLDL and 13HODE are elevated in ACM patients. However, unfortunately no data is provided to show that oxLDL is actually taken up in cardiomyocytes of ACM patients. The oxidized oxLDL uptake studies are confined to isolated C-MSCs. In addition, evidence that oxLDL is taken up into cardiomyocytes by CD36 is missing. In experiments in which CD36 is silenced in C-MSCs no data is provided as to whether this modified oxLDL uptake.

The study also does not provide strong evidence that oxLDL is a source of 13HODE. Regardless, although plasma 13HODE levels are increased in ACM patients. No evidence is provided to show that oxLDL is releasing 13HODE to activate PPAR γ . In addition, adding 13HODE to c-MSCs from ACM patients does not increase PPAR or lipid accumulation in ACM C-MSCs (Figure 3B-C).

The link between oxidative stress and internalization of oxLDL is also weak. No strong data is presented to support the contention that increased oxidative stress is promoting oxLDL internalization (see Synopsis figure).

Specific Comments:

- 1) Figures should contain individual data points.
- 2) Figure 1C: The relationship between oxLDL and scar tissue is not very convincing. The positive relationship in the graph relies heavily on outlier data point.
- 3) Figure 2D: CD36 is not significantly increased in ACM cells. It is implied that CD36 is increased. However, the lack of increase in ACM cells does not fit with the data in Figure 1D? Are these cells representative of ventricles?
- 4) Fig. 2D: This data does not support the concept that CD36 is responsible for the increase in Dil oxLDL uptake (Figure 2F).
- 5) Figure 3B: 13HODE did not increase ORO or PPAR γ in ACM. This is unlike oxLDL treatment. How do the authors explain this?
- 6) Figure 3: Where is the data showing the oxLDL effects on PPAR γ and CD36? This is a logical experiment that is missing.
- 7) Figure 3: NAC decreased PPAR γ and CD36 in ACM. However, it had no real effect specifically related to 13HODE.
- 8) Figure 4: What does siRNA for CD36 do to ORO response to 13HODE? No data is provided that examines this.
- 9) Online Figure 5: lipids, MDA and CD36 expression are normal in Pkp2 \pm mice? Why?
- 10) Figure 1: What is relationship between 13HODE and scar tissue? Why is only the oxLDL relationship shown?

Milan, 06/04/2021

REBUTTAL LETTER

As agreed with the Editor, we send, as a *de novo* submission, a second version of the manuscript "Oxidized-LDL/CD36/PPAR γ circuitry is a trigger of adipogenesis in Arrhythmogenic Cardiomyopathy" (EMM-2020-12531). The new title is "**Oxidized LDL-dependent pathway as new pathogenic trigger in Arrhythmogenic Cardiomyopathy.**"

In this new version, we expanded our cohort, genotyped each patient, and enriched the manuscript with new analyses on patients' clinical data, which considerably strengthen our message, reinforcing the translational outlook of our results. Moreover, the present version of the manuscript includes new experimental data, which address the referees' concerns.

In the manuscript our hypothesis was consistently demonstrated by a multilevel approach, spanning from cell models, mouse models and patients, providing insights at molecular, cellular, tissue and systemic levels.

We further showed that oxLDL plasma levels constitute an indicator of patient disease severity, which may be leveraged as precision medicine tool or to evaluate response to therapy.

Importantly, the therapeutic effectiveness of our approach is readily exploitable in the clinical scenario of Arrhythmogenic Cardiomyopathy, to date lacking a specific preventive therapy.

We provide below a point-by-point response to all the Reviewers' issues. We feel that the manuscript is now significantly improved.

Referee #2 (Comments on Novelty/Model System for Author):

The findings presented here extend the well described involvement of PPAR signaling for other metabolic related disease to ACM. Further, the use of statins to attenuate incidences of ventricular arrhythmias has been previously described. The model systems used for experimentation are appropriate.

Referee #2 (Remarks for Author):

The manuscript by Sommariva et al. describes an oxLDL/CD36/PPAR circuitry axis in which elevations in oxLDL cellular uptake, driven by CD36 up-regulate PPAR expression and activity. In a feed forward circuit, the authors describe such phenomenon as a potential mechanism contributing to lipid accumulation in ACM affected cells. Moreover, the authors translate their in vitro findings using a genetic mouse model and demonstrate the utility of cholesterol lowering pharmacology to alleviate the disease phenotype. The findings reported here are a great advancement to understanding the (patho)biology of ACM. However, there are several limitations and concerns that warrant further consideration.

We thank the reviewer for these comments and for the recognition of the impact of our results on ACM pathobiology. The described biological axis is pharmacologically-targetable, and the fact that statins attenuate the incidences of ventricular arrhythmias in other cardiac diseases, as correctly stated by the Reviewer, supports the proposal of their use in ACM.

As detailed below, we have addressed all the issues raised, which we feel have strengthen the manuscript.

Major concerns:

1) The authors should explain why they observe significantly more ROS and lipid peroxidation in C-MSCs in ACM vs HC cells, but no difference in cellular oxidant defense mechanisms. Moreover, the authors should also describe why the addition of the antioxidant NAC decreased lipid accumulation (as concluded to be driven by oxidant stress) in ACM C-MSCs, yet those cells (based on the authors' data) do not appear to have any defects in cellular antioxidant defense.

As correctly pointed out, we did not detect defects in antioxidant defences, but we found an increase of oxidative stress in our cell model and in the ventricular tissue of ACM patients. As already shown in other cardiovascular conditions (doi: 10.1155/2020/5732956), this unbalance is due to the saturation of the antioxidant capacity, which impedes ACM C-MSC to face the increased levels of oxidative stress, as shown below. Indeed, by potentiating antioxidant defences with NAC, oxidative stress levels and lipid peroxidation are counteracted

(10.1371/journal.pone.0147858; 10.3390/nu11122850; 10.1097/MD.000000000013087; **Rebuttal Figure 1:** analysis of MDA treated cells).

Rebuttal Figure 1

We also demonstrated that NAC reduces the effects of oxidised lipids in their ability to activate PPAR γ . The described balance is depicted by the following model (**Rebuttal Figure 2**).

Rebuttal Figure 2

In addition, NAC has been shown to exert a direct effect on the transcriptional profile of the described circuitry, including CD36 expression (doi:10.1007/s11033-011-1062-1) and PPAR γ (10.1016/j.biopha.2011.04.020). Discussion has been improved accordingly, in lines 368-371.

Mechanistically, we speculate that the increased levels of oxidative stress may be related to mitochondrial defects (doi:10.3389/fphys.2019.01496). This issue deserves further investigations, as stated in the Discussion paragraph of our manuscript (lines 358-359). Moreover, the effects of physical exercise should be taken into an account, since many ACM patients are athletes or very active people, and exercise is known to affect redox balance (10.1152/jappl.1978.45.6.927; 10.1007/BF00690898; 10.1113/JP270646). In addition, an interdependence between oxidative stress and inflammation, well known ACM feature, has been described (doi:10.1155/2016/5698931).

2) The authors should make a better attempt to elucidate a mechanism by which relatives carrying the ACM mutation do not display the ACM phenotype. It is also not clear whether oxLDL is the consequence of or the cause of the ACM phenotype.

The explanation of the high phenotypic variability among ACM mutation carriers, raised by the Reviewer, is a key clinical need in ACM. Conceivably, there must be something beyond ACM causative genes. The central hypothesis of our work is that oxLDL levels are a disease co-factor that modulates ACM phenotype. This hypothesis is supported by the following data:

- The observation that oxLDL levels are low in the family members of ACM patients, carriers of the same mutation as their affected relatives, excludes the possibility that oxLDL is a direct consequence of the main causative mutation. Under this view, the ACM-causative mutation confers the susceptibility to ACM phenotype. We have addressed this point by silencing *PKP2* in control cells, which show an increased susceptibility to oxLDL (Appendix; Figure S4). This predisposition needs to be triggered by hyperactivation of PPAR γ , in order to reach full expressivity.
- Speculations on possible reasons of oxLDL different levels are discussed in lines 331-340 of the manuscript, and include secondary genetic traits, oxidative status due to exercise or mitochondrial defects or inflammation, diet and lifestyle.
- Our novel clinical data show that oxLDL is associated to ACM phenotype progression.

Our experimental data support the hypothesis that oxLDL is a **mediator of an additional causative mechanism**, beyond the predisposing genetic defect. This is supported by the following data:

- oxLDL treatment worsens C-MSC lipid accumulation;
- by silencing CD36 levels, we obtained lower oxLDL internalization (new data Figure 5D), and a reduction of PPAR γ -mediated lipid accumulation; on the other hand, by antagonizing PPAR γ we reduced oxLDL internalization. Therefore, the proposed circuitry is causally involved;
- by increasing oxLDL levels with a high fat diet, we exacerbated the ACM phenotypes in the *Pkp2* mouse model.

We cannot exclude that also disease severity may increase oxLDL levels in patients, as an example by boosting ROS production and inflammation, or by other unknown mechanisms. In any case, our data, including the newly generated ones (Figure 2), demonstrated the advantage of low levels of oxLDL in terms of phenotypic expression of the disease (see Discussion paragraph; lines 388-391).

3) Given the susceptibility of this disease model to cardiac arrhythmias, the authors need to characterize the electrical activity in their mouse model with programmed electrical stimulation.

Our study was designed to assess the cardiac substrate changes and the consequent functional defect. Thus, unfortunately, we were not cleared by the Italian Competent Authority for the suggested additional experiments. However, we obtained new data showing a higher arrhythmic risk in ACM patients with circulating oxLDL above the cutoff value of 86ng/ml. Notably, this observation is in line with the reported effects of oxLDL on arrhythmic risk (doi: 10.3390/antiox9121213; 10.1016/j.cardiores.2004.12.009; 10.1016/j.cardiores.2005.11.019). We believe that these clinical observations may overcome the issue of cardiac arrhythmias in the mouse model.

4) The mechanistic work dissecting the oxLDL/CD36/PPAR circuitry could be better described. For example, the difference in CD36 expression is marginal and it would be expected that silencing CD36 expression would preclude entry of oxLDL into the cell. Moreover, the mechanistic work describing the biological function of PPAR should be presented as a main figure, as this is central to the work reported here.

As suggested by the Reviewer, we provided a more complete description of the mechanistic process, through additional experiments. This data set is detailed in the new Figure 4.

We performed an oxLDL internalization assay in ACM cells silenced for CD36, which resulted in lower oxLDL uptake, even if CD36 protein mean reduction was in the order of 30%, as correctly pointed out by the Reviewer. It has, however, to be highlighted in this regard, that this reduction is statistically significant and correlates with a reduction in PPAR γ levels.

Furthermore, we added the analysis of CD36 protein expression following PPAR γ antagonism, which resulted decreased, as expected. The whole experiment has now been moved to Figure 4.

5) Figure 2D - The authors should represent an uncut gel, or where splice and with more representative lanes. It's unclear how the ratio of POI/GAPDH is \sim 1.0 when clearly the density of the POIs are much less than GAPDH. Importantly, it does not look apparent that any difference in the POIs exist between groups. Given the authors attempt to correlate increased CD36/PPAR expression in ACM affected cells, this should be significant attention.

Moreover, if the authors used numerical transformation for control/experimental normalization, this should be described in the methods section.

As suggested, we modified the representative images of WB, to better represent the differential expression of PPAR γ and CD36 between ACM and HC.

As correctly interpreted by the Reviewer, we used the mean of HC cell values as reference, expressing it as 1, in order to better visualize the fold difference between ACM and HC. This numerical transformation allows to have comparable scales of values that do not depend on the unit of measurement, not modifying the correlations. As for the correlation in Figure 5C, we carefully checked, as suggested by the Reviewer, that the correlation was maintained even using raw data, as shown below (Rebuttal Figure 3A). Also with another method of numerical transformation (standardization; Rebuttal Figure 3B) we obtained an identical correlation goodness index ($R^2=0.756$). The correlation coefficient is $R=0.869$. We have better described our normalization method in the Appendix (lanes 247-248; 449-450).

Rebuttal Figure 3.

6) Given that HC hiPSC-CMs are immune to lipid accumulation under AC + PPAR agonistic conditions, whereas ACM hiPSC-CMs are not might suggest dysfunctional lipid metabolism. The authors should explore this using mitochondrial functional assays (Seahorse).

Actually, a predisposition to lipid accumulation was already reported in CM derived from iPSC lacking PKP2. A dysfunctional lipid metabolism and mitochondrial activity in iPSC-CM carrying *PKP2* mutation has been described by the group of HSV Chen (doi: 10.1038/nature11799) and by S Hu (doi: 10.1126/scitranslmed.aay8329).

To provide a different perspective in our model, we measured CD36 expression and functionality upon PPAR γ agonism in our ACM iPSC-CM. Results demonstrated that CD36 protein levels and oxLDL internalization are increased upon rosiglitazone stimulus (Appendix; Figure S5).

7) It's important to not only describe the accumulation of lipids in ACM affected cells, but also how it relates to the pathological consequence to the disease.

This point is indeed essential for the clinical relevance of our findings and needs to be addressed at different levels.

- 1) Lipid accumulation in the heart is known to be deleterious not only because it contributes to the presence of non-conductive and non-contractile tissue, but adipocyte-secreted factors have been demonstrated to negatively influence CM contractility (doi: 10.3389/fphys.2018.01752; 10.1161/CIRCRESAHA.109.200501; 10.1097/MCO.0b013e328358be7b; 10.1038/ijo.2014.193).

- 2) Accordingly, in our murine model the presence of fat associates with initial RV systolic impairment, as happens in the first phases of ACM patients' cardiac dysfunction.
- 3) The new clinical data shown in Figure 2 indicate that the subgroup of patients with oxLDL plasma levels above the cut-off show a worse ACM phenotype in terms of fat accumulation, right/biventricular dysfunction and arrhythmic burden.

These observations have been better clarified in the Discussion paragraph.

8) The authors may consider using their mouse model to identify cell populations in the heart responsible for fatty tissue deposition.

As suggested by the Reviewer, we confirmed in the ACM mouse model that those cells undergoing adipogenic differentiation are of mesenchymal origin (Appendix; Figure S7), as demonstrated in ACM patients (doi: 10.1093/eurheartj/ehv579).

Minor concerns:

1) The manuscript should be professionally proofread for English grammar.

As suggested, the manuscript has been proofread for English grammar.

2) Individual data points for bar graphs should be presented.

We now changed all graphs adding single data points.

3) Figure 1C - with the exception of a couple potential outliers, it doesn't appear that there is any correlation between scar and oxLDL. It's interesting that $p=0.03$. The authors should report R squared with regression statistics.

We agree with the Reviewer that the evidence provided with this analysis is not enough conclusive. We then decided to remove it. However, we have refined the substrate remodeling read-out providing a new analysis of ventricular fat accumulation at MRI. In particular, we quantified ventricular fat by evaluating type 2 chemical shift /Indian ink artifact at SSFP sequences for every patient of our cohort who underwent MRI at our hospital. We found that the oxLDL cutoff discriminated two subpopulation of ACM patients with a statistically significant difference of fat accumulation (Figure 2B). We believe that this analysis is more relevant for our study, since LGE marks both fibrosis and adipogenesis, while chemical shift identifies fat, the main disease phenotype linked to oxLDL.

4) Figures 1D,E - Normalizing to nuclei is not appropriate for this measurement. There are apparent differences in nuclei counts between the two images, which largely skews the data (especially in figure 1D. Further, there appears to be autofluorescence. Please describe how this was corrected for data normalization. A better method for MDA histology be colorimetric microscopy using agents such as DAB.

We understand the concern of the Reviewer since the representative image show a clear difference in the nuclei count between ACM and HC. However, the quantification was made on 15 images each sample, not just the one shown, and the total count of nuclei in the 15 fields was similar (e.g. ACM $n=1471$ vs. HC $n=1442$). We elected to show an image with adipocytes, which is more representative of ACM. In any case, to address Reviewer concern, we double checked the difference between ACM and HC using a normalization based on the tissue area. As expected, the ACM samples remained significantly different vs. HC (Rebuttal Figure 4).

Rebuttal Figure 4.

To reassure the Reviewer, immunofluorescence and acquisitions were performed at the same time and with the same acquisition/exposure parameters, in order to make them comparable for a relative quantification. A negative control, in which secondary antibody without the primary was used to check for autofluorescence and/or aspecific staining, was performed for each antibody. The software AxioVision Rel. 4.8. only quantify the signal above a threshold decided by the operator. Negative control was used to determine the threshold. Therefore, we did not quantify autofluorescence. This is now better explained in the “Methods” section, Appendix, lines 122-125.

The choice of fluorescent antibodies rather than DAB-based ones was made to allow quantification of the fluorescence intensity, which is dependent on the quantity of antibody linked to the signal, rather than a mere quantification of the area of signal, which is what our equipment allows with DAB (doi: 10.1016/S0002-9440(10)63984-3). Fluorescent antibody method is accepted for MDA quantification (doi: 10.1186/2051-5960-1-61; 10.1038/s41467-020-17915-w; 10.1038/s41598-021-82481-0).

5) Figure 3C - Lanes on the representative gel should be labeled.

Thank you, we have now labelled the lines of the Western blot

6) Reporting of non-significant data as a trend should be minimized.

We check results and discussion and avoided it.

7) My comment #4 should be extended to all figures using that methodology.

According to the observations above in response to the reviewer issue, we did not change the quantification method. However, in order to convince the reviewer that the quantification is reliable, we validated it by western blot analyses. Results of the validation are reported in the Appendix (Figure S9).

Referee #3 (Comments on Novelty/Model System for Author):

This paper addressed an interesting topic. The authors hypothesize that an oxLDL/CD36/PPAR γ circuitry triggers adipogenesis in arrhythmogenic cardiomyopathy. However, I believe key data is missing to support this hypothesis, particularly evidence that oxLDL is actually been taken up into the cardiomyocytes of ACM patients. Other experimental data needed to support the conclusions is also missing. These and other concerns are discussed in detail in the "Comments to Authors".

Referee #3 (Remarks for Author):

This study addressed the hypothesis that oxLDL uptake into cardiomyocytes, via CD36, activates PPAR γ and triggers adipogenesis in arrhythmogenic cardiomyopathy (ACM). ACM patients showed higher levels of oxLDL and 13HODE compared to healthy controls. In ACM patient cardiac-mesenchymal stromal cells (C-MSCs) oxLDL internalization through CD36 resulted in PPAR γ up-regulation and lipid accumulation. In *Pkp2* +/- mice a high fat diet increased cardiac adipogenesis, which was prevented by atorvastatin-treatment. It is concluded that altered oxidized lipid metabolism and oxidative stress increase oxLDL bioavailability, which is internalized by C-MSC by CD36 receptor, thus

acting through 13HODE-mediated PPAR γ activation to promote cardiac adipogenic differentiation.

General Comments:

This paper addresses the intriguing hypothesis that cardiomyocyte oxLDL internalization via CD36 can promote adipogenesis in ACM.

We thank the Reviewer for the interest in the topic. We demonstrated that oxLDL internalization via CD36 promotes adipogenesis in ACM stromal cells, which are the main effector of cardiac adipogenesis. To reinforce the hypothesis, mainly focused on the stromal cell compartment, we also showed that cardiomyocytes derived from iPSC, during PPAR γ agonism, increase CD36 expression and function (Appendix; Figure S5).

The authors do show that plasma levels of oxLDL and 13HODE are elevated in ACM patients. However, unfortunately no data is provided to show that oxLDL is actually taken up in cardiomyocytes of ACM patients. The oxidized oxLDL uptake studies are confined to isolated C-MSCs.

As requested by the Reviewer, we now provide evidence that DiI-oxLDL is internalised in iPSC-derived cardiomyocytes, and in particular at a higher extent in ACM than in HC cells and further upon PPAR γ stimulation (Appendix; Figure S5C).

In addition, evidence that oxLDL is taken by CD36 is missing. In experiments in which CD36 is silenced in C-MSCs no data is provided as to whether this modified oxLDL uptake.

We thank the reviewer for giving us the chance to complete the studies on the mechanism responsible for the trigger of adipogenesis (Figure 5). As requested, we have provided evidence that oxLDL entry in C-MSC is mediated by CD36. Indeed, when CD36 is silenced, less DiI-oxLDL gets internalised in the cells (Figure 5D).

The study also does not provide strong evidence that oxLDL is a source of 13HODE. Regardless, although plasma 13HODE levels are increased in ACM patients. No evidence is provided to show that oxLDL is releasing 13HODE to activate PPAR γ .

Robust evidence is available in the literature that during oxidation of LDL, different oxidation products are generated, among which free fatty acid derivatives, such as 13HODE (doi: 10.1089/ars.2009.2733; PMID: 2373954; 10.1016/s0092-8674(00)81574-3). Indeed, among LDL lipids, arachidonic and linoleic acid are the major targets of oxidizing agents. These fatty acids are released from sn-2 position of oxidized phospholipids by different phospholipases and are oxidized by various lipoxygenases to their hydroxy derivatives. For 13HODE, further reduction to 13-oxo-ODE, a potent ligand for PPAR- γ , is catalysed by the 13HODE dehydrogenase (10.1177/2042018810375656). Several methods are utilized to oxidize LDL, with different results in terms of production of hydroxyacids. Non-enzymatic reactions are commonly used for the *in vitro* oxidation of LDL. Lenz et al (PMID: 2373954) were among the first to report that oxidation of LDL for 24 h in the presence of 5 μ M Cu²⁺ decreases linoleate and arachidonate. In the final preparation, 9HODE and 13HODE accounted for 67% of the linoleate consumed. Similar results were also reported by Esterbauer et al. (10.1016/S0022-2275(20)38678-8) and by Coutant et al (10.4049/jimmunol.172.1.54), where 9HODE and 13HODE accounted for 60% of all lipid peroxidation products found in oxLDL. In our experimental conditions we produced oxLDL following what documented by these and several other publications (/10.1021/bi700225a; 10.1016/S0021-9150(99)00456-6; 10.1359/jbmr.1999.14.12.2067). For our oxLDL preparations, after LDL dialysis and copper-induced oxidation, we verified the outcome of the procedure by assessing the migration of oxLDL vs native ones by agarose gel and by measuring the relative disappearance of PUFAs (mainly linoleic and arachidonic acids). In these conditions, we therefore expect a yield of HODEs similar to that found by the abovementioned authors.

In addition, the notion that oxidized fatty acid derivatives from linoleic and arachidonic acids (among which 13HODE), contained in oxLDL, elicit PPAR γ activation has been demonstrated long ago (doi: 10.1016/S0092-8674(00)81574-3; 10.1073/pnas.94.9.4318; 10.1007/s11745-014-3954-z; 10.1038/22572; 10.3390/ijms19051529; PMID: 10787429; 10.1007/s11010-005-5873-z). In the last decades, a plethora of PPAR γ -mediated effects of oxLDL

have been demonstrated to be dependent on 9HODE and 13HODE, such as induction of ALPB expression (10.1016/s0021-9150(02)00305-2), stimulation of the expression of MCP-3 (10.1016/j.bbrc.2004.08.178), suppression of CCR2 expression (10.1172/JCI10052), VEGF production in macrophages and endothelial cells (10.1161/01.atv.21.4.560). Limor et al (10.1038/ajh.2007.39) documented that 13HODE increases the expression of PPAR γ mRNA in vascular smooth muscle cells, suggesting a novel amplification cycle in which PPAR γ activation induces production of 12- and 15-LO-derived metabolites, which in turn feedback to upregulate PPAR γ 's own expression. Notably, as already highlighted in our paper, Nagy et al. elegantly showed (further documented by Fischer), that exposure of monocytes/macrophages to oxLDL provokes activation and expression of PPAR γ , via 9HODE and 13HODE (10.1016/s0092-8674(00)81574-3; 10.4049/jimmunol.168.6.2828). Jostarndt et al also demonstrated that 13HODE increases CD36 and FABP4 expression by activating PPAR γ . Interestingly, FABP4 expression in THP1 cells is increased by 9HODE, 13HODE, and its overexpression leads to marked increase in lipid accumulation (10.1016/s0021-9150(02)00305-2). Altogether, these evidence point to a conserved and robust effect elicited by 13HODE in bridging signalling between oxLDL and PPAR γ activation.

In addition, adding 13HODE to c-MSCs from ACM patients does not increase PPAR or lipid accumulation in ACM C-MSCs (Figure 3B-C).

Figure 4B and C clearly show that 13HODE treatment do increase lipid accumulation, PPAR γ and CD36 expression. The red bars show comparisons intra-ACM group, and asterisks represent statistical significance between adipogenic medium (AM) versus AM +13HODE experimental condition, which are significantly different for ORO accumulation ($p < 0.05$), PPAR γ protein levels ($p < 0.05$) and CD36 protein levels ($p < 0.01$).

The link between oxidative stress and internalization of oxLDL is also weak. No strong data is presented to support the contention that increased oxidative stress is promoting oxLDL internalization (see Synopsis figure).

We apologise for being unclear on this message. We never claimed that oxidative stress promotes oxLDL cellular uptake. Oxidative stress is instead the likely cause of the oxidation of LDL. Indeed, an impaired redox balance can induce oxidation of different proteins and complexes, including LDL (doi: 10.1186/s12944-021-01435-7; 10.1007/s11883-017-0678-6). oxLDL can be internalized by cells which express the scavenger receptor CD36, such as C-MSC. We now better describe the hypothesis in the Synopsis Figure legend (lines 633-639).

Specific Comments:

1) Figures should contain individual data points.

As requested we now have changed all graphs to show individual data points.

2) Figure 1C: The relationship between oxLDL and scar tissue is not very convincing. The positive relationship in the graph relies heavily on outlier data point.

We agree with the Reviewer that the evidence provided with scar tissue analysis is not fully convincing. To better address the ACM tissue remodeling phenotype, we have quantified ventricular fat at MRI by evaluating type 2 chemical shift (Indian ink artifact) at SSFP sequences in our patients. The new clinical data (Figure 2) show that oxLDL levels stratify the ACM population according to the severity of the disease, even as for ventricular fat accumulation (Figure 2B). We believe that this readout is more relevant than LGE, because chemical shift identifies fat, which is the main disease phenotype linked to oxLDL.

3) Figure 2D: CD36 is not significantly increased in ACM cells. It is implied that CD36 is increased. However, the lack of increase in ACM cells does not fit with the data in Figure 1D? Are these cells representative of ventricles?

We understand the concern of the reviewer. In Figure 1D, we performed a whole ventricle analysis of pathological tissue, which includes cardiomyocytes and different cell types other than C-MSC. Therefore, the relative contribution of each cell type to CD36 levels is not defined. As for cultured ACM C-MSC, mimicking *in vitro* the pathological environment through adipogenic medium, an increase of CD36 expression is observed (Figure 3E).

Therefore, these primary cells are representative of the diseased ACM ventricle when exposed to pathogenetic stimulus (10.1093/eurheartj/ehv579).

4) Fig. 2D: This data does not support the concept that CD36 is responsible for the increase in Dil oxLDL uptake (Figure 2F).

In addition to the concepts expressed in response to point 3), we demonstrated CD36 importance in cell oxLDL uptake by means of the new internalization assay performed in CD36 silenced C-MSK (Figure 5D). As described in the Results section (lines 256-258), reduced CD36 levels determined lower oxLDL internalization.

5) Figure 3B: 13HODE did not increase ORO or PPAR γ in ACM. This is unlike oxLDL treatment. How do the authors explain this?

As discussed above, Figure 4A, B and C clearly show that 13HODE and oxLDL treatments do increase lipid accumulation, PPAR γ and CD36 expression.

6) Figure 3: Where is the data showing the oxLDL effects on PPAR γ and CD36? This is a logical experiment that is missing.

We now added to the new Figure 4 the analysis of PPAR γ and CD36 expression of in HC and ACM C-MSK after oxLDL treatment (Figure 4A).

7) Figure 3: NAC decreased PPAR γ and CD36 in ACM. However, it had no real effect specifically related to 13HODE.

The effects of NAC in reducing lipid accumulation, CD36 and PPAR γ expression are evident both when used in combination with 13HODE and when used alone (Figure 4B and C). We acknowledge this in the Results section (lines 233-244). We now have improved the discussion (lines 368-371), by adding the consideration that NAC may be able to reduce even the basal level of oxidative stress, being likely beneficial on ACM lipid accumulation independently on additional oxidised cofactors.

8) Figure 4: What does siRNA for CD36 do to ORO response to 13HODE? No data is provided that examines this.

Data is provided in response to oxLDL, which is the main focus of the paper. Moreover, CD36 is a specific receptor of oxLDL (PMID: 7685021), thus the experiment was designed to test CD36-mediated oxLDL uptake and effects. 13HODE is known to be internalized not only through CD36, but also by membrane diffusion (doi: 10.1023/A:1020542220599; 10.1074/jbc.M011623200; 10.1016/j.plefa.2016.05.005). This could have hampered the effective demonstration of the role of CD36 in the hypothesized circuitry. The analyses of 13HODE performed in Figure 4 was meant to specify the effects of one of the active components of oxLDL.

9) Online Figure 5: lipids, MDA and CD36 expression are normal in Pkp2 \pm mice? Why?

The whole hypothesis is based on the fact that the ACM mutation alone do not cause sufficient oxidative stress to activate the proposed pathway. However, through other known mechanisms directly dependent on desmosomal dysfunction (e.g. Wnt and Hippo pathways), PPAR γ is more expressed in ACM mouse hearts than in WT ones, but not enough to provoke the initiation of the circuitry and cardiac adipogenesis. By inducing oxidative stress and LDL oxidation by high fat diet, PPAR γ becomes overactivated above a threshold, which leads to adipogenesis and higher transcription of CD36. The observations of patient data lead to the same conclusions. The individuals which are carriers of an ACM mutation but not exposed to oxidative stress, keep low their oxLDL levels, and, in parallel, disease signs. The symptomatic ACM patients are ACM mutation carriers who instead are exposed to oxidative stress. Their higher oxLDL levels worsen PPAR γ , CD36 expression, adipogenesis, and clinical phenotypes. As for patients, we could not follow up disease progression, whereas the *in vivo* model pathogenic evolution demonstrated our hypothesis.

10) Figure 1: What is relationship between 13HODE and scar tissue? Why is only the oxLDL relationship shown?

As pointed out earlier, 13HODE is only one of the active components in oxLDL activating PPAR γ (10.3390/ijms19051529). Therefore, we elected oxLDL as stratifying parameter (Figure 2), refining the substrate analysis with fat quantification.

7th May 2021

Dear Dr. Sommariva,

Thank you for the resubmission of your manuscript to EMBO Molecular Medicine, and please accept my apologies for the delay in getting back to you, which is due to the fact that one referee needed more time to complete his/her review. We have now received feedback from the three reviewers who agreed to evaluate your manuscript (referees #1 and #2 had already reviewed the first version of your manuscript). As you will see from the reports below, the referees acknowledge the interest of the study and are overall supporting publication of your work pending appropriate revisions.

Addressing the reviewers' concerns in full will be necessary for further considering the manuscript in our journal, and acceptance of the manuscript will entail another round of review. Acceptance or rejection of the manuscript will depend on the completeness of your responses included in the next, final version of the manuscript.

When submitting your revised manuscript, please carefully review the instructions that follow below. Failure to include requested items will delay the evaluation of your revision:

2) Individual production quality figure files as .eps, .tif, .jpg (one file per figure). Please make sure to provide high resolution figures.

3) A .docx formatted letter INCLUDING the reviewers' reports and your detailed point-by-point responses to their comments. As part of the EMBO Press transparent editorial process, the point-by-point response is part of the Review Process File (RPF), which will be published alongside your paper.

4) A complete author checklist, which you can download from our author guidelines (<https://www.embopress.org/page/journal/17574684/authorguide#submissionofrevisions>). Please insert information in the checklist that is also reflected in the manuscript. The completed author checklist will also be part of the RPF.

6) Before submitting your revision, primary datasets produced in this study need to be deposited in an appropriate public database (see <https://www.embopress.org/page/journal/17574684/authorguide#dataavailability>). The accession numbers and database should be listed in a formal "Data Availability" section (placed after Materials & Method). Please note that the Data Availability Section is restricted to new primary data that are part of this study.

7) We would also encourage you to include the source data for figure panels that show essential data. Numerical data should be provided as individual .xls or .csv files (including a tab describing the data). For blots or microscopy, uncropped images should be submitted (using a zip archive if multiple images need to be supplied for one panel). Additional information on source data and instruction on how to label the files are available at .

8) Our journal encourages inclusion of *data citations in the reference list* to directly cite datasets that were re-used and obtained from public databases. Data citations in the article text are distinct from normal bibliographical citations and should directly link to the database records from which the data can be accessed. In the main text, data citations are formatted as follows: "Data ref: Smith et al, 2001" or "Data ref: NCBI Sequence Read Archive PRJNA342805, 2017". In the Reference list, data citations must be labeled with "[DATASET]". A data reference must provide the database name, accession number/identifiers and a resolvable link to the landing page from which the data can be accessed at the end of the reference. Further instructions are available at .

9) We replaced Supplementary Information with Expanded View (EV) Figures and Tables that are collapsible/expandable online. A maximum of 5 EV Figures can be typeset. EV Figures should be cited as 'Figure EV1, Figure EV2' etc... in the text and their respective legends should be included in the main text after the legends of regular figures.

- Additional Tables/Datasets should be labeled and referred to as Table EV1, Dataset EV1, etc. Legends have to be provided in a separate tab in case of .xls files. Alternatively, the legend can be supplied as a separate text file (README) and zipped together with the Table/Dataset file. See detailed instructions here:

10) For more information: There is space at the end of each article to list relevant web links for further consultation by our readers. Could you identify some relevant ones and provide such information as well? Some examples are patient associations, relevant databases, OMIM/proteins/genes links, author's websites, etc...

11) Every published paper now includes a 'Synopsis' to further enhance discoverability. Synopses are displayed on the journal webpage and are freely accessible to all readers. They include a short stand first (maximum of 300 characters, including space) as well as 2-5 one-sentences bullet points that summarizes the paper. Please write the bullet points to summarize the key NEW findings. They should be designed to be complementary to the abstract - i.e. not repeat the same text. We encourage inclusion of key acronyms and quantitative information (maximum of 30 words / bullet point). Please use the passive voice. Please attach these in a separate file or send them by email, we will incorporate them accordingly.

Please also suggest a striking image or visual abstract to illustrate your article as a png file 550 px-

wide x 400-px high.

12) As part of the EMBO Publications transparent editorial process initiative (see our Editorial at <http://embomolmed.embopress.org/content/2/9/329>), EMBO Molecular Medicine will publish online a Review Process File (RPF) to accompany accepted manuscripts.

In the event of acceptance, this file will be published in conjunction with your paper and will include the anonymous referee reports, your point-by-point response and all pertinent correspondence relating to the manuscript. Let us know whether you agree with the publication of the RPF and as here, if you want to remove or not any figures from it prior to publication.

I look forward to receiving your revised manuscript.

Yours sincerely,

Lise Roth

Lise Roth, PhD
Editor
EMBO Molecular Medicine

To submit your manuscript, please follow this link:

in not Available

Photos 400-800 DPI

*Additional important information regarding figures and illustrations can be found at <https://bit.ly/EMBOPressFigurePreparationGuideline>

***** Reviewer's comments *****

Referee #1 (Comments on Novelty/Model System for Author):

The authors use a number of important model systems to address their hypothesis.

Referee #1 (Remarks for Author):

The authors have done a good job of addressing the concerns that I raise in the initial review of this manuscript. This includes the addition of important new data that supports their hypothesis.

Referee #2 (Remarks for Author):

The revised manuscript (albeit submitted as de novo) is greatly improved. The authors have taken considerable efforts to address my comprehensive list of concerns. There is, however, one concern remaining that should be addressed, see below.

1) While the authors have done an outstanding job at provided better in vitro data to support their conclusions, they still have not directly tied this with any electrical abnormalities. Their mouse model serves as an appropriate model system for such experimentation. If, for some reason, the authors cannot obtain regulatory approval for such in vivo experimentation (which is rather peculiar), ex vivo experimentation such as multielectrode array, isochronal mapping, Langendorf ECG prep, etc. is a great alternative.

Referee #3 (Comments on Novelty/Model System for Author):

This is a well performed, thorough study that could potentially have important clinical implications for patients with ACM or patients carrying an ACM related mutation.

The samples and models use are suitable and all support the posed hypotheses. I am very excited about the findings. While the data might be in line with expectations, this is actually the first study the shows the link between circulating levels of oxLDL and cardiac adipogenic remodeling during ACM.

Referee #3 (Remarks for Author):

This manuscript by Sommariva et al. describes the influence of circulating levels of oxLDL on ACM disease penetrance. The authors show that ACM patients have higher circulating levels of oxLDL which correlates with a higher expression level of CD36 (its receptor), MDA (lipid peroxidation marker) in tissue samples from ACM patients and that there is a direct link between circulating levels of oxLDL and cardiac remodeling and function.

In vitro the authors showed that c-MSCs from ACM patients show higher levels of oxidative stress and expression of CD36 and PPARgamma and are able to internalize more oxLDL.

These findings are validated in vivo as PKP2 mutant mice on HFD show more cardiac lipid accumulation and dysfunction than their corresponding controls. Cholesterol lowering drug are able

to rescue this phenotype, further underscoring the influence of oxidative stress in triggering the adipogenic remodeling.

This is a clearly presented and important study that poses the intriguing hypothesis that oxLDL internalization via CD36 stimulates adipogenesis and thereby contributes to ACM pathogenesis. The conclusions are supported by an impressive set of human data and studies performed in relevant in vitro models and in vivo models.

Some clarifications could be added to further strengthen the story:

- The authors should better explain the c-MSC model and why they are using these cells for their studies? In general the authors could try to be a bit more clear about their experimental set up in the text.
- Where are the c-MSCs coming from? Which patients are used to collect cells from?
- In authors mention in the methods section they use iPS-CMs, but it is unclear where these cells are used or how they have been characterized.
- The authors mention an experiment in which they use an siRNA against PKP2 to lower PKP2 levels and show that this causes susceptibility to oxLDL. Does this mean that the susceptibility is only present in patients showing a lower level of PKP2? How does this match with the patients samples used in Figure 1 and 2?

Referee #1 (Comments on Novelty/Model System for Author):

The authors use a number of important model systems to address their hypothesis.

Referee #1 (Remarks for Author):

The authors have done a good job of addressing the concerns that I raise in the initial review of this manuscript. This includes the addition of important new data that supports their hypothesis.

We thank the Reviewer for recognizing the multilevel approach to the demonstration of our hypothesis and our effort to comprehensively answer to the important concern raised.

Referee #2 (Remarks for Author):

The revised manuscript (albeit submitted as *de novo*) is greatly improved. The authors have taken considerable efforts to address my comprehensive list of concerns. There is, however, one concern remaining that should be addressed, see below.

1) While the authors have done an outstanding job at provided better *in vitro* data to support their conclusions, they still have not directly tied this with any electrical abnormalities. Their mouse model serves as an appropriate model system for such experimentation. If, for some reason, the authors cannot obtain regulatory approval for such *in vivo* experimentation (which is rather peculiar), *ex vivo* experimentation such as multielectrode array, isochronal mapping, Langendorf ECG prep, etc. is a great alternative.

We thank the Reviewer for recognising the effort made to provide adequate answer to the interesting issues raised in the first round of revisions.

The Reviewer correctly points out that we have not shown the electrical phenotype in our mouse model. We like to highlight, however, that we have shown that higher levels of oxLDL define an ACM patient population with greater arrhythmic risk in terms of major arrhythmic events.

On the experimental side, we regret to admit that obtaining clearance for additional mouse experiments, including *ex-vivo* testing, given the tight restrictions in our country, would need a “*de novo*” regulatory process that does not allow a due time rebuttal. However, we thank the reviewer because this question prompted us to analyse surface electrocardiogram (ECG) tracings acquired in mice during echography. Interestingly, while the ECG of WT and *Pkp2*^{+/-} animals fed a chow diet were similar, we found a striking difference in the traces of *Pkp2*^{+/-} animal fed high fat diet (HFD). In particular, i) QRSp duration, defined as in doi: 10.14814/phy2.12639, was higher in *Pkp2*^{+/-} HFD than in the other conditions, ii) terminal activation duration (TAD; J wave) was prolonged and fragmented, iii) QRS amplitude was reduced (Appendix **FigureS11A**, lines 595-603). All these features, suggestive of conduction slowing in the right ventricle, have

been described in ACM patients (doi: 10.1161/CIRCEP.108.832519; 10.1016/j.hrthm.2008.07.012; 10.1161/JAHA.118.009855), and are predictors of arrhythmic risk (doi: 10.1111/jce.12202; 10.2459/JCM.0b013e32834bed0a).

In addition, QRS widening has been reported for *N271S-Dsg2* ACM transgenic mice (doi: 10.1093/cvr/cvs219; 10.1084/jem.20090641) and in the same *Pkp2*^{+/-} model we used, under stress conditions like flecainide induction (doi: 10.1093/cvr/cvs218) or trans-aortic constriction (doi: 10.3390/ijms20174076).

In addition, since it is well known that ventricular conduction velocity (reflected by QRS duration; doi: 10.1016/j.pbiomolbio.2017.08.003) and arrhythmic burden are related to connexin-43 (CX43) expression and localization (doi: 10.1016/j.bbamem.2011.07.039), we analysed mouse cardiac tissue by CX43 immunofluorescence. We found that Cx43 was less expressed and mislocalized in *Pkp2*^{+/-} HFD mice (Appendix **FigureS11B**, lines 603-608).

We believe that these data demonstrate that mouse electric activity is impaired by the combination of genotype and HFD, and reflects substrate and functional changes we have observed (Figure6).

Interestingly, the aforementioned electrical-related abnormalities were prevented by the use of atorvastatin (Appendix **FigureS12**, lines 609-619), in agreement with what, once again, we have observed at functional and tissue level (Figure7).

Referee #3 (Comments on Novelty/Model System for Author):

This is a well performed, thorough study that could potentially have important clinical implications for patients with ACM or patients carrying an ACM related mutation.

The samples and models use are suitable and all support the posed hypotheses. I am very excited about the findings. While the data might be in line with expectations, this is actually the first study that shows the link between circulating levels of oxLDL and cardiac adipogenic remodeling during ACM.

Referee #3 (Remarks for Author):

This manuscript by Sommariva et al. describes the influence of circulating levels of oxLDL on ACM disease penetrance. The authors show that ACM patients have higher circulating levels of oxLDL which correlates with a higher expression level of CD36 (its receptor), MDA (lipid peroxidation marker) in tissue samples from ACM patients and that there is a direct link between circulating levels of oxLDL and cardiac remodeling and function.

In vitro the authors showed that c-MSCs from ACM patients show higher levels of oxidative stress and expression of CD36 and PPAR γ and are able to internalize more oxLDL.

These findings are validated *in vivo* as PKP2 mutant mice on HFD show more cardiac lipid accumulation and dysfunction than their corresponding controls. Cholesterol lowering drug are able to rescue this phenotype, further underscoring the influence of oxidative stress in triggering the adipogenic remodeling.

This is a clearly presented and important study that poses the intriguing hypothesis that oxLDL internalization via CD36 stimulates adipogenesis and thereby contributes to ACM pathogenesis. The conclusions are supported by an impressive set of human data and studies performed in relevant *in vitro* models and *in vivo* models.

We thank the reviewer for appreciating our work and foreseeing its possible exploitation in the clinical scenario.

Some clarifications could be added to further strengthen the story:

- The authors should better explain the c-MSC model and why they are using these cells for their studies? In general, the authors could try to be a bit more clear about their experimental set up in the text.
- Where are the c-MSCs coming from? Which patients are used to collect cells from?

C-MSC are primary cells obtained by the processes of digestion and plastic adherence from a biopsy of the right ventricle of patients and controls. The process of obtainment was previously described in the paper of Pilato et al. (doi: 10.3791/57263), while their role in ACM is explained in Sommariva et al. (doi: 10.1093/eurheartj/ehv579). In the present manuscript, this cell model was mentioned in the introduction (lines 100-101), in the manuscript Method section (lines 440-450), and in the Appendix (lines 110-115 and 133-145).

ACM patients and healthy donors were the sources of ACM C-MSC, and their use in each experiment of the manuscript is detailed in the Appendix TableS6.

According to the Reviewer advice, we have better described C-MSC in the introduction (lines 100-106) and slightly improved the experimental setup description in line 126.

- In authors mention in the methods section they use iPS-CMs, but it is unclear where these cells are use or how they have been characterized.

Despite we believe that the cells mainly responsible for adipocyte accumulation in ACM are C-MSC (doi: 10.1093/eurheartj/ehv579), we are aware that cardiomyocyte can undergo lipogenesis, which may dysregulate both their contractility and their electrical function. Therefore, most of our *in vitro* experiments are performed on C-MSC, but a validation of the hypothesis is present also using a human cardiomyocyte cell model (Appendix, FigureS5, described in lines 551-563).

The used cell lines have been produced from blood cells of a patient carrying a mutation in *PKP2* (p.N346Lfs*12) and a healthy control of the same family, by episomal procedure, as detailed in the Appendix methods (lines 302-310). The characterization of the lines were performed as previously described (doi: 10.1016/j.scr.2017.12.012). Briefly, episomal vector persistence was excluded by PCR, pluripotency confirmed by checking the expression of the pluripotency markers SSEA-4, OCT3/4, TRA-1-80,

SOX-2, TRA-1-60 and alkaline phosphatase, and the karyotype analysis was performed to exclude gross chromosomal rearrangements. The detailed characterization of the used iPSC lines is not provided here as it is the object of a separate manuscript in preparation for Stem Cell Research (Lab Resources).

Cardiomyocyte differentiation is detailed in the Appendix methods (lines 312-326).

- The authors mention an experiment in which they use an siRNA against PKP2 to lower PKP2 levels and show that this causes susceptibility to oxLDL. Does this mean that the susceptibility is only present in patients showing a lower level of PKP2? How does this match with the patients samples use in Figure 1 and 2?

We thank the reviewer for the interesting question. Our interpretation of this result is that plakophilin haploinsufficiency can be one of the mechanisms (probably not the only) to induce susceptibility to oxLDL. Our data and data from other groups showed that PKP2 reduction is quite common in ACM patients, even in those who did not carry *PKP2* mutations (doi: 10.1093/eurheartj/ehv579; doi: 10.1161/CIRCRESAHA.114.302810).

28th Jun 2021

Dear Dr. Sommariva,

Thank you for the submission of your revised manuscript to EMBO Molecular Medicine. We have now received the reports from the two referees who reviewed your study. As you will see, they are supportive of publication, and I am therefore pleased to inform you that we will be able to accept your manuscript once the following editorial points will be addressed:

1/ Main manuscript text:

- Please answer/correct the changes in track changes mode suggested by our data editors in the main manuscript file labelled 'Related manuscript file'. Please use this file for any further modification.
- Please remove the changes in red, and only keep in track changes mode the new modifications.
- Please modify the sentences highlighted in pink in the attached screenshots so as to avoid similarities with previously published work (beginning of the introduction and of The Paper Explained).
- Please remove the abbreviation list, and instead incorporate the abbreviations in the manuscript text.
- Please introduce a space before a parenthesis with references.
- Material and methods: we note that your detailed methods are displayed in the Appendix. Please move most of the methods to the main manuscript file. Moreover, please include the full statement that the experiments conformed to the principles set out in the WMA Declaration of Helsinki and the Department of Health and Human Services Belmont Report.
- Data availability section: Thank you for providing a Data Availability section. Please provide a direct link to access the data and note that the data have to be publicly available before acceptance of the manuscript. Please adjust the sections 19 and 20 of the checklist accordingly.
- Authors' contribution: please differentiate between Michela Casella/ Mattia Chiesa/ Marina Camera.
- Please merge the funding information with the Acknowledgements.
- Please update the reference format so as to have 10 authors listed before et al.

2/ Figures and Appendix:

- Please indicate in the main and appendix figures or in their legends the exact $n=$ and exact $p=$ values, not a range, along with the statistical test used, including for non-significant p -values. Some people found that to keep the figures clear, providing a supplemental table in the appendix with all exact p -values was preferable. You are welcome to do this if you want to.
- Please remove the figures from the main manuscript file. The Appendix figures should be in the Appendix only (not uploaded as separate files).
- Please add a table of content to your appendix file, and rename the figures and tables "Appendix Figure S1" etc. and "Appendix Table S1" etc.
- Please rename the File S1 as "Dataset EV1" and add a title and a legend.
- Table S6 and S7 are not referenced in the main text, please update the callouts accordingly

3/ We would also encourage you to include the source data for figure panels that show essential data. Numerical data should be provided as individual .xls or .csv files (including a tab describing the data). For blots or microscopy, uncropped images should be submitted (using a zip archive if

multiple images need to be supplied for one panel). Additional information on source data and instruction on how to label the files are available at

In particular, please provide the raw data for figure 4C.

4/ As part of the EMBO Publications transparent editorial process initiative (see our Editorial at <http://embomolmed.embopress.org/content/2/9/329>), EMBO Molecular Medicine will publish online a Review Process File (RPF) to accompany accepted manuscripts.

This file will be published in conjunction with your paper and will include the anonymous referee reports, your point-by-point response and all pertinent correspondence relating to the manuscript. Let us know whether you agree with the publication of the RPF and as here, if you want to remove or not any figures from it prior to publication.

I look forward to receiving your revised manuscript.

With kind regards,

Lise Roth

Lise Roth, PhD
Editor
EMBO Molecular Medicine

To submit your manuscript, please follow this link:

in ot Availa le

Photos 400-800 DPI

*Additional important information regarding figures and illustrations can be found at <https://bit.ly/EMBOPressFigurePreparationGuideline>

The system will prompt you to fill in your funding and payment information. This will allow Wiley to send you a quote for the article processing charge (APC) in case of acceptance. This quote takes into account any reduction or fee waivers that you may be eligible for. Authors do not need to pay any fees before their manuscript is accepted and transferred to our publisher.

***** Reviewer's comments *****

Referee #2 (Comments on Novelty/Model System for Author):

While the authors still have not performed detailed EP analyses, they have provided new data to answer the research question. It is now my opinion this manuscript is suitable for publication.

Referee #2 (Remarks for Author):

The authors have addressed my concerns to the best of their ability citing time constraints. The manuscript is greatly improved.

Referee #3 (Remarks for Author):

This is an important study and there are no further comments from my side

The authors performed the requested editorial changes.

6th Jul 2021

Dear Dr. Sommariva,

Thank you for submitting the revised files. We are now pleased to inform you that your manuscript is accepted for publication in EMBO Molecular Medicine!

Please note that the section F of the checklist needs to be corrected (no data deposited in a public repository), therefore please make the necessary changes and send us via email the corrected file as soon as possible. Your manuscript will then be sent to our publisher to be included in the next available issue of EMBO Molecular Medicine.

Congratulations on your interesting work!

With kind regards,

Lise Roth

Lise Roth, Ph.D
Editor
EMBO Molecular Medicine

Follow us on Twitter @EmboMolMed
Sign up for eTOCs at embopress.org/alertsfeeds

*** ** IMPORTANT INFORMATION ** **

SPEED OF PUBLICATION

The journal aims for rapid publication of papers, using the advance online publication "Early View" to expedite the process: A properly copy-edited and formatted version will be published as "Early View" after the proofs have been corrected. Please help the Editors and publisher avoid delays by providing e-mail address(es), telephone and fax numbers at which author(s) can be contacted.

Should you be planning a Press Release on your article, please get in contact with embomolmed@wiley.com as early as possible, in order to coordinate publication and release dates.

LICENSE AND PAYMENT:

All articles published in EMBO Molecular Medicine are fully open access: immediately and freely available to read, download and share.

EMBO Molecular Medicine charges an article processing charge (APC) to cover the publication costs. You, as the corresponding author for this manuscript, should have already received a quote

with the article processing fee separately. Please let us know in case this quote has not been received.

Once your article is at Wiley for editorial production you will receive an email from Wiley's Author Services system, which will ask you to log in and will present you with the publication license form for completion. Within the same system the publication fee can be paid by credit card, an invoice, pro forma invoice or purchase order can be requested.

Payment of the publication charge and the signed Open Access Agreement form must be received before the article can be published online.

PROOFS

You will receive the proofs by e-mail approximately 2 weeks after all relevant files have been sent to our Production Office. Please return them within 48 hours and if there should be any problems, please contact the production office at embopressproduction@wiley.com.

Please inform us if there is likely to be any difficulty in reaching you at the above address at that time. Failure to meet our deadlines may result in a delay of publication.

All further communications concerning your paper proofs should quote reference number EMM-2021-14365-V3 and be directed to the production office at embopressproduction@wiley.com.

Corresponding Author Name: Elena Sommariva
Journal Submitted to: EMBO Molecular Medicine
Manuscript Number: EMM-2021-14365